# PixelVLA: Advancing Pixel-level Understanding in Vision-Language-Action Model

**Wenqi Liang**[1,2], **Gan Sun**[2†], **Yao He**[2], **Jiahua Dong**[3], **Suyan Dai**[2], **Ivan Laptev**[3],
**Salman Khan**[3,4], **Yang Cong**[2]

[1] University of Trento
[2] School of Automation Science and Engineering, South China University of Technology
[3] Mohamed bin Zayed University of Artificial Intelligence
[4] Australian National University

{liangwenqi0123, sungan1412, heyao0293, dongjiahua1995,
congyang81}@gmail.com, {ivan.laptev, salman.khan}@mbzuai.ac.ae
† Corresponding Author

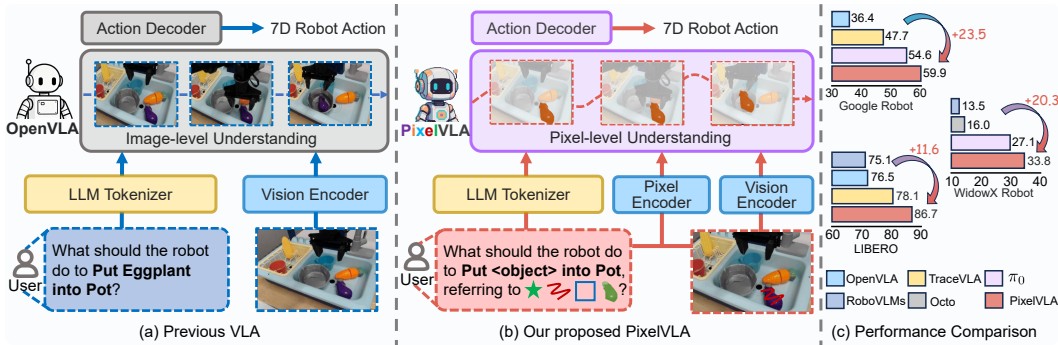

Figure 1: We introduce PixelVLA, a vision–language–action (VLA) model designed for pixel-level reasoning and multimodal prompting. Unlike prior VLA models (a), which primarily rely on image-level understanding for manipulation and depend solely on textual instructions, PixelVLA (b) advances beyond these limitations by enabling fine-grained pixel-level comprehension and supporting both textual and visual prompts. This paradigm effectively enhances spatial precision and expands human–robot interaction, leading to superior performance (c) compared to baseline methods.

## ABSTRACT

Vision-Language-Action models (VLAs) are emerging as powerful tools for learning generalizable visuomotor control policies. However, current VLAs are mostly trained on large-scale image–text–action data and remain limited in two key ways: (i) they struggle with pixel-level scene understanding, and (ii) they rely heavily on textual prompts, which reduces their flexibility in real-world settings. To address these challenges, we introduce PixelVLA, the first VLA model designed to support both pixel-level reasoning and multimodal prompting with text and visual inputs. Our approach is built on a new visuomotor instruction tuning framework that integrates a multiscale pixel-aware encoder with a visual prompt-aware encoder. To train PixelVLA effectively, we further propose a two-stage automated annotation pipeline that generates Pixel-160K, a large-scale dataset with pixel-level annotations derived from existing robot data. Experiments on three standard VLA benchmarks and two VLA model variants show that PixelVLA improves manipulation success rates by $10.1\% \sim 28.7\%$ over Open-VLA, while requiring only $1.5\%$ of its pretraining cost. These results demonstrate that PixelVLA can be integrated into existing VLAs to enable more accurate, efficient, and versatile robot control in complex environments. Project page: https://wenqiliang.github.io/PixelVLA/.

# 1 INTRODUCTION

Traditional robotic policy learning methods (Brohan et al. (2022); Liang et al. (2024); Chi et al. (2023); Dong et al. (2026)) rely heavily on task-specific demonstration datasets (James et al. (2020); Liu et al. (2023a)), which limits their ability to generalize to out-of-distribution (OOD) tasks. In contrast, vision-language-action models (VLAs) (Brohan et al. (2023); Kim et al. (2024); Black et al. (2024)) leverage large-scale robot datasets together with pre-trained vision-language models (VLMs), achieving stronger generalization and instruction-following capabilities. For example, RT-2 (Brohan et al. (2023)) integrates internet-scale VLMs with robotic control, enabling semantic reasoning and manipulation of novel objects. Similarly, OpenVLA (Kim et al. (2024)) leverages Prismatic VLM (Karamcheti et al. (2024)) as backbone to conduct large-scale training on the OXE dataset (O'Neill et al. (2024)), leading to significant improvement in OOD generalization.

Despite recent progress, as shown in Fig. 1 (a), most VLAs (Kim et al. (2024); Han et al. (2026); Shi et al. (2025)) inherit from VLMs that process observations only at the image level, lacking fine-grained pixel-level understanding. This gap limits spatial reasoning and weakens OOD generalization. In contrast, pixel-level comprehension has already been successfully validated in VLMs (Ren et al. (2024); Zhang et al. (2024b)) and enables precise object perception and richer spatial awareness, which are key for robust manipulation in diverse environments. The second limitation lies in prompting. Most VLAs depend solely on textual instructions, which overlook subtle visual cues and constrain spatial awareness (Ranasinghe et al. (2024)) and multimodal human–robot interaction (Jiang et al. (2023); Zheng et al. (2024); Li et al. (2026)). To explore visual prompting in VLAs, TraceVLA (Zheng et al. (2024)) improves spatial-temporal awareness with visual traces, and LLaRA (Li et al. (2025b)) encodes object locations within textual prompts to enhance region-level understanding. Nevertheless, these approaches still face challenges in achieving fine-grained pixel-level understanding and effectively integrating diverse multimodal prompts (*e.g.*, points, lines, regions, masks) (Wu et al. (2024b); Dong et al. (2024)).

Inspired by the successful visual instruction tuning in VLMs (Liu et al. (2023b); Karamcheti et al. (2024)), we introduce a novel visuomotor instruction tuning framework to train our VLA models. This framework is designed to significantly enhance the pixel-level understanding capabilities of VLAs and empower them to effectively process multimodal visuomotor control prompts. However, current robotic datasets (O'Neill et al. (2024); Khazatsky et al. (2024)) lack multimodal prompts and pixel-level annotations. Meanwhile, directly employing existing VLMs and open-set segmentation models (Karamcheti et al. (2024); Liu et al. (2024a)) to extract visual prompts and pixel-level annotations proves to be ineffective. This is due to a significant domain gap between their pre-training data and robotic data, as well as the cluttered and low-quality nature of robotic images.

To tackle the above challenges, as presented in Fig. 1 (b), we introduce PixelVLA in this paper, the first vision-language-action model that achieves both **pixel-level understanding** and **multimodal prompting**. The model architecture of PixelVLA comprises a pre-trained VLMs as backbone, a visual prompt-aware encoder, a multiscale pixel-aware encoder and a continuous action decoder. Specifically, in PixelVLA, we introduce a lightweight visual prompt-aware encoder to process the diverse visual prompts (*e.g.*, points, lines, regions, masks). Subsequently, a novel multiscale pixel-aware encoder is designed to generate pixel-aware embeddings to inject pixel-level understanding into VLAs. Furthermore, we develop a continuous action representation decoder that leverages pixel-level understanding to capture fine-grained action details based on the hidden states of VLMs.

To address the challenge of synthesizing high-quality multimodal prompts and pixel-level annotations from cluttered, low-quality robot observations, we propose a two-stage automated annotation pipeline to create a pixel-annotated visuomotor instruction tuning dataset, namely **Pixel-160K**. Concretely, our two-stage automated annotation pipeline comprises a gripper-aware region proposal stage followed by a multimodal object segmentation stage. In the first stage, a video segmentation model is employed to localize the robot gripper and generate preliminary region proposals for target objects. Subsequently, the second stage leverages a large language model (LLM) and an open-vocabulary segmentation model to predict pixel-level annotations and produce multimodal prompts from these region proposals. Thereafter, we train PixelVLA using the proposed visuomotor instruction tuning framework, which incorporates a continuous action training stage and a pixel-level understanding enhancement stage. To evaluate the effectiveness of PixelVLA, we integrate its architecture and visuomotor instruction-tuning procedure into two widely adopted VLAs:

OpenVLA (O'Neill et al. (2024)) and $\pi_0$ (Black et al. (2024)). Extensive evaluations on three VLA benchmarks demonstrate that PixelVLA advances current VLAs to achieve superior performance in zero-shot manipulation tasks and adaptation to new robot setups, while requiring only $1.5\%$ of the pretraining computation of OpenVLA.

The main contributions of this paper are listed below:

- We present PixelVLA, a novel vision-language-action model enabling pixel-level understanding while supporting both textual and visual prompts. In PixelVLA, we introduce a lightweight visual prompt-aware encoder to process diverse visual prompts, a novel multi-scale pixel-aware encoder for pixel-level understanding injection, and a continuous action decoder to generate robotic action.

- We design a novel two-stage automated annotation pipeline to effectively create a pixel-level visuomotor instruction tuning dataset from the publicly available robot datasets, called Pixel-160K, where the pipeline comprises the gripper-aware region proposal stage and the multimodal object segmentation stage.

- We introduce a novel visuomotor instruction tuning framework for training PixelVLA, comprising a continuous action training stage and a pixel-level understanding enhancement stage. Extensive evaluations on three benchmarks and two VLA model variants show that PixelVLA improves performance of current VLAs with relatively low training cost.

## 2 RELATED WORK

**Vision-Language-Action Models**. Vision-language-action models (VLAs) (Team et al. (2025); Brohan et al. (2023); Black et al. (2024); Ding et al. (2024); Fan et al. (2025)) have propelled robotic manipulation forward by endowing robots with the ability to understand and execute language-based instructions in diverse visual environments. Trained on numerous robot episodes, OpenVLA (Kim et al. (2024)) enables zero-shot control and adaptation for various robots. Building on the foundational capabilities of OpenVLA, various approaches have been proposed to advance robotic manipulation, such as SpatialVLA (Qu et al. (2025)) and ECoT (Zawalski et al. (2024)). Most prior VLAs focus on innovations in visual processing for robotic manipulation, such as introducing visual chain-of-thought (CoT) reasoning mechanisms for visual planning (Zhao et al. (2025)). Nevertheless, they primarily process visual information at the image level, lacking the ability to perform detailed pixel-level visual processing required for precise robotic manipulation.

**Visual Prompting in VLMs**. Visual prompting methods (Zhang et al. (2023); Ren et al. (2024); Wu et al. (2024a); Zhang et al. (2024b)) have recently emerged as a complementary paradigm to textual prompting, allowing models to accept more fine-grained supervision in the form of region-level (Guo et al. (2024)) and even pixel-level instructions (Ma et al. (2024); Rasheed et al. (2024)) over multimodal inputs. Shikra (Chen et al. (2023)) extends MLLMs with a simple vision–encoder–LLM architecture that treats spatial coordinates as natural language tokens. Ferret (You et al. (2023)) enhances region-level grounding in MLLMs through a hybrid region representation and a spatial-aware visual sampler that supports diverse region inputs, while Ferret-v2 (Zhang et al. (2024a)) further introduces any-resolution grounding and multi-granularity visual encoding, leading to improved fine-grained visual understanding and localization over prior MLLMs. LocVLM (Ranasinghe et al. (2024)) proposes an image-space coordinate–based instruction fine-tuning framework for MLLMs that explores coordinate representations and spatial objectives to inject spatial awareness. However, despite these advances, robust pixel-level understanding in VLA frameworks remains challenging, especially when aligning fine-grained spatial cues with continuous, high-precision action control.

**Visual Instruction Tuning in VLAs**. Visual Instruction Tuning (Zhu et al. (2023); Liu et al. (2023b); Rasheed et al. (2024)) is generally divided into two steps, which are modality alignment and instruction optimization, respectively. This strategy also serves as the core paradigm for realizing multimodal capabilities in VLAs (Li et al. (2025b), Zheng et al. (2024); Zawalski et al. (2024)). For example, TraceVLA (Zheng et al. (2024)) introduces visual trace prompting to enhance spatial-temporal awareness in VLAs. In contrast, LLaRA (Li et al. (2025b)) reformulates the robot action policy as visuo-textual conversations through visuomotor instruction tuning and RoVI (Li et al. (2025c)) develops an object-centric visual instruction paradigm with symbolic sketches. However, to address various visuomotor control challenges, adapting visual instruction tuning for VLAs remains a major constraint.

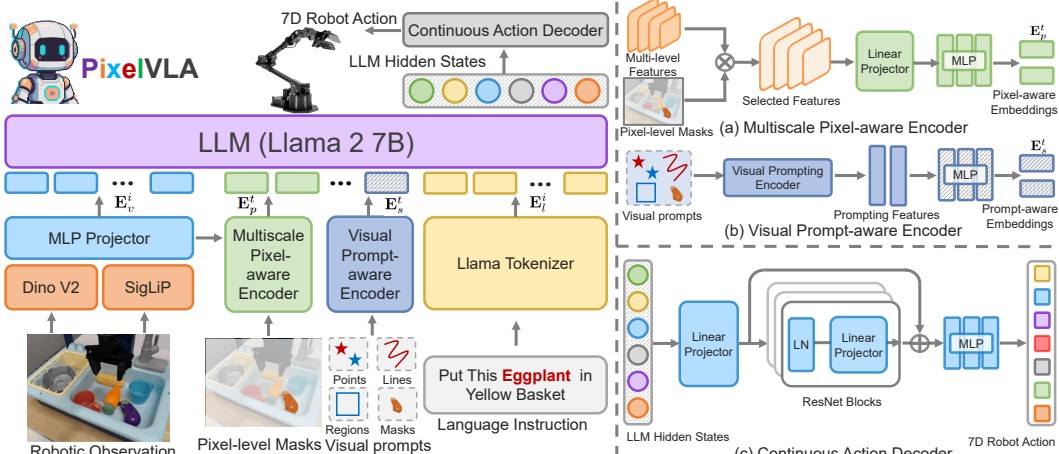

Figure 2: Overview of the PixelVLA architecture. The model integrates three novel components: (1) a *visual prompt-aware encoder* for processing input diverse visual prompts; (2) a *multiscale pixel-aware encoder* that injects pixel-level information into token embeddings; and (3) a *continuous action decoder* to predict 7D robot actions. PixelVLA enhances fine-grained pixel-level spatial understanding and multimodal prompt responsiveness, enabling more precise manipulation policies in visually complex scenarios.

## 3   PROBLEM DEFINITION: VISUOMOTOR INSTRUCTION TUNING

Inspired by the effectiveness of visual instruction tuning in VLMs (Liu et al. (2023b); Rasheed et al. (2024); Wu et al. (2024a); Kang et al. (2025); Li et al. (2025a)), we aim to adapt a similar process for VLAs to tackle diverse visuomotor control challenges (*e.g.*, various multimodal prompts) and achieve pixel-level understanding. Similar to LLaRA (Li et al. (2025b)), we formalize this paradigm as Visuomotor Instruction Tuning. Specifically, following OpenVLA (Kim et al. (2024)), given a series of image observations $\mathbf{X} = \{\mathbf{x}^t \in \mathbb{R}^{H \times W \times 3}\}_{t=1}^{T}$ and a language instruction $\mathbf{L}$, the VLA model $\mathcal{F}_\theta(\cdot)$ can generate a series of robotic actions $\mathbf{A} = \{\mathbf{a}^t \in \mathbb{R}^7\}_{t=1}^{T}$, *i.e.*, $\mathbf{a}^t = \mathcal{F}_\theta(\mathbf{x}^t, \mathbf{L})$. For an episode of length $T$, the likelihood of successfully completing the task through an action sequence $\mathbf{A}$ can be calculated as:

$$p(\mathbf{A}|\mathbf{X}, \mathbf{L}) = \prod_{t=1}^{T} p_\theta(\mathbf{a}^t|\mathbf{x}^t, \mathbf{L}), \qquad (1)$$

where $T$ denotes the length of timestep in an episode, $\theta$ represents the parameters of VLA model $\mathcal{F}_\theta(\cdot)$ and $p_\theta$ denotes the likelihood of generating action $\mathbf{a}^t$ by the VLA model $\mathcal{F}_\theta(\cdot)$. However, this robotic action generation process fails to accommodate various visual prompts and achieve fine-grained pixel-level understanding. To address these challenges, we here introduce a novel visuomotor instruction tuning framework that reformulates robotic action generation as $\mathbf{a}^t = \mathcal{F}_\theta(\mathbf{x}^t, \mathbf{p}^t, \mathbf{L}, \mathbf{V})$ and reformulate the likelihood in Eq. (1) for an episode of length $T$ as:

$$p(\mathbf{A}|\mathbf{X}, \mathbf{P}, \mathbf{L}, \mathbf{V}) = \prod_{t=1}^{T} p_\theta(\mathbf{a}^t|\mathbf{x}^t, \mathbf{p}^t, \mathbf{L}, \mathbf{V}), \qquad (2)$$

where $\mathbf{P} = \{\mathbf{p}^t \in \mathbb{R}^{H \times W}\}_{t=1}^{T}$, $\mathbf{p}^t$ represents the pixel-aware mask input, and $\mathbf{V}$ denotes the diverse visual prompts (*e.g.*, points, lines, regions, masks).

## 4   THE PROPOSED METHOD

As illustrated in Fig. 2, we present the architecture of the proposed PixelVLA to achieve pixel-level understanding and accommodate both textual and visual prompts. Specifically, PixelVLA integrates a novel multiscale pixel-aware encoder (Sec. 4.1) that infuses pixel-level understanding into VLAs through tokenized representations, a visual prompt-aware encoder for handling diverse visual prompts (Sec. 4.1), and a continuous action decoder (Sec. 4.1) for accurate robotic action

prediction. In addition, an automated annotation generation pipeline and a pixel-annotated visuo-motor instruction tuning dataset Pixel-160K are presented in Sec. 4.2. Subsequently, we introduce the proposed visuomotor instruction tuning procedure for training PixelVLA in Sec. 4.3.

## 4.1 PixelVLA Architecture

Current VLAs (Black et al. (2024); Kim et al. (2024); Wang et al. (2026b;a)) are typically pre-trained on large-scale image-instruction-action robotic datasets (O'Neill et al. (2024); Wu et al. (2024c)). Architecturally built upon VLMs, these models process single or multi-view images along with textual instructions. However, this foundation inherently restricts their ability to achieve pixel-level understanding or respond to detailed visual prompts, resulting in constraining VLAs for spatial comprehension and object perception.

To address these architectural constraints, as illustrated in Fig. 2, we present a novel VLA model, namely PixelVLA. Specifically, PixelVLA integrates four main parts: (1) a **vision encoder** and **MLP projector** for visual embedding extraction, (2) a **visual prompt-aware encoder** and a **multiscale pixel-aware encoder** for accommodating visual prompts and pixel-level understanding injection, (3) a **LLM backbone** and (4) a **continuous action decoder** for non-discrete robot action prediction. Following OpenVLA (Kim et al. (2024)), we preliminarily build our PixelVLA on Prismatic-7B VLM (Karamcheti et al. (2024)), where a Llama 2-7B (Touvron et al. (2023)) is employed as LLM backbone. The vision encoder of PixelVLA consists of pre-trained DinoV2 (Oquab et al. (2023)) and SigLIP (Zhai et al. (2023)) models, and a lightweight 2-layer MLP projector is utilized to map the output features of the vision encoder into the input space of LLM.

**Multiscale Pixel-aware Encoder**. To extract pixel-level information from multiscale image features and encode the spatial positional information of visual prompts, we propose a multiscale pixel-aware encoder designed to generate both pixel-aware embeddings and prompt-aware embeddings. Specifically, as described in Eq. (2), for each training sample $\{\mathbf{x}^0, \mathbf{p}^0, \mathbf{L}^0, \mathbf{V}^0\}$ drawn from Pixel-160K, PixelVLA first encodes the image observation $\mathbf{x}^0 \in \mathbb{R}^{H \times W \times 3}$ with the SigLIP vision encoder to obtain multi-level visual features $\mathbf{F}_v^0 = \{\mathbf{f}_v^{0,i} \in \mathbb{R}^{H_i \times W_i \times D_i}\}_{i=1}^L$, where $L$ denotes the number of selected feature levels. As illustrated in Fig. 2 (a), the multiscale pixel-aware encoder leverages the features $\mathbf{F}_v^0$ and a pixel-aware mask input $\mathbf{p}^0 \in \mathbb{R}^{H \times W}$ to compute the pixel-aware embeddings $\mathbf{E}_p^0 \in \mathbb{R}^{N_p \times D}$. Here, $N_p$ is the length of pixel-aware embeddings and $D$ denotes the feature dimension of LLM. Specifically, the pixel-aware embeddings $\mathbf{E}_p^0$ can be computed as follows:

$$\mathbf{E}_p^0 = \mathrm{MLP}(\sum_{i=1}^L \Gamma^i(\mathbf{f}_p^{0,i})), \quad \mathbf{f}_p^{0,i} = \frac{\mathbf{p}^0 \cdot \mathbf{f}_v^{0,i}}{|\mathbf{p}^0|}, \tag{3}$$

where $\mathrm{MLP}(\cdot)$ is a multilayer perceptron (MLP) layer and $\Gamma^i(\cdot)$ denotes the linear projection in the $i$-th linear projector. Supervised by the action prediction loss, PixelVLA learns to associate the pixel-level information encoded in these pixel-aware embeddings $\mathbf{E}_p^0$ with action generation, thereby enhancing the VLA backbone with pixel-level understanding.

**Visual Prompt-aware Encoder**. As shown in Fig. 2(b), we adopt a lightweight prompt encoder similar to that in SAM (Kirillov et al. (2023)) and integrate it into PixelVLA as the visual prompting encoder. Concretely, the user-provided prompts $\mathbf{V}^0 \in \mathbb{R}^{H \times W}$ are first converted into continuous positional embeddings based on their normalized image coordinates, and then combined with learned prompt-type embeddings to produce prompt features $\mathbf{F}_s^0 \in \mathbb{R}^{N_s \times D_s}$, where $N_s$ is the embedding length and $D_s$ is the feature dimension. These features $\mathbf{F}_s^0$ are further transformed by an MLP to obtain the final prompt-aware embeddings $\mathbf{E}_s^0 \in \mathbb{R}^{N_s \times D}$. Since each embedding is explicitly tied to a specific location or region in the image via its coordinate-based positional embedding, the spatial positional information of the visual prompts is preserved throughout the encoding process.

**Continuous Action Decoder**. Most existing VLA models (Kim et al. (2024); Zheng et al. (2024); Li et al. (2024b)) adapt autoregressive generation to predict sequential action tokens based on the pre-trained VLM backbone. In contrast, following $\pi_0$ (Black et al. (2024)), we develop a continuous action decoder that directly predicts continuous action representations, leveraging pixel-level understanding to capture fine-grained action details. Specifically, as illustrated in Fig. 2 (b), the hidden states $\mathbf{F}^t \in \mathbb{R}^{N_s \times D}$ from the last layer of LLM backbone are sequentially processed by a linear projector, $N_r$ ResNet blocks and a MLP projector to obtain the actions $\mathbf{A} \in \mathbb{R}^{N_c \times 7}$. Here,

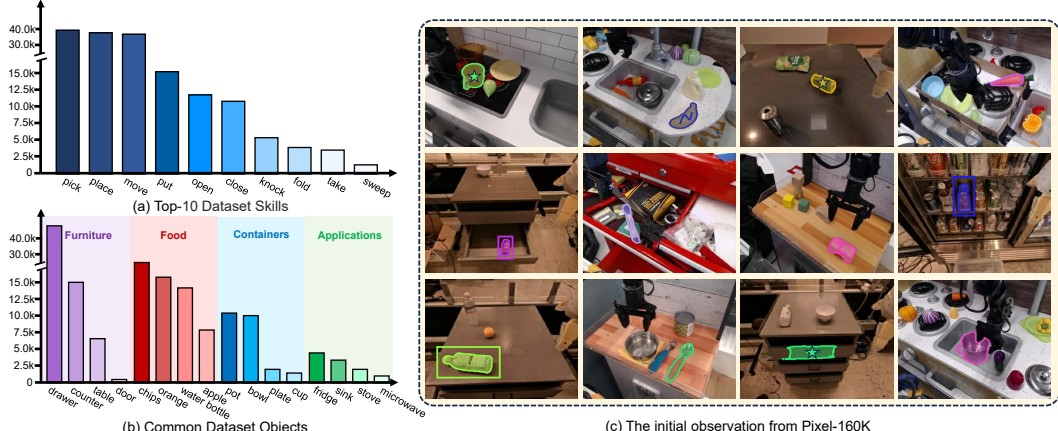

Figure 3: Overview of the Pixel-160K Dataset.

$N_s$ denotes the sequence length of the LLM backbone, while $N_c$ represents the chunk size used in action chunking (Zhao et al. (2023)). In this way, we can effectively preserve the pixel-level understanding learned by the pre-trained VLM backbone while enabling the continuous action decoder to incorporate these features directly into the continuous action prediction. The resulting continuous actions are then used to compute an L1 regression loss that supervises the training process.

## 4.2 VISUOMOTOR TUNING DATA GENERATION

In this section, we introduce Pixel-160K as shown in Fig. 3, a visuomotor instruction tuning dataset comprising image-text-action triplets with visual prompts and mask annotations, containing approximately 160K manipulation episodes to encourage VLAs for fine-grained pixel-level understanding. Specifically, to address the challenge of cluttered and low-quality robot observations in robot datasets, we propose an automated annotation pipeline containing a gripper-aware region proposal stage and a multimodal object segmentation stage. This pipeline enables the effective generation of visual prompts and mask annotations for each episode using the publicly available Fractal dataset (Brohan et al. (2022)) and Bridge v2 dataset (Walke et al. (2023)).

**Gripper-aware Region Proposal Stage.** Given a sequence of observations $\{\mathbf{x}_\eta^1, \mathbf{x}_\eta^2, \ldots, \mathbf{x}_\eta^{N_\eta}\}$ from the $\eta$-th episode, the first gripper-close state in the episode as $G_\eta \in \{1, 2, \ldots, N_\eta\}$ and the corresponding observation as $\mathbf{x}_\eta^{G_\eta} \in \{\mathbf{x}_\eta^1, \mathbf{x}_\eta^2, \ldots, \mathbf{x}_\eta^{N_\eta}\}$. Here, $N_\eta$ represents the length of the $\eta$-th episode. Sequentially, we can select a series of gripper-close state observations $\{\mathbf{x}_1^{G_1}, \mathbf{x}_2^{G_2}, \ldots, \mathbf{x}_{N_e}^{G_{N_e}}\}$ from the whole dataset, where $N_e$ denotes the number of total episodes in the dataset. Furthermore, we assume $\{\mathbf{x}_1^{G_1}, \mathbf{x}_2^{G_2}, \ldots, \mathbf{x}_{N_e}^{G_{N_e}}\}$ as a discrete video and apply SAM 2 (Ravi et al. (2024)) to generate $N_e$ gripper masks. Then, we compute the minimal axis-aligned bounding boxes enclosing these masks, uniformly enlarge each by a fixed margin to capture local context and reduce detection noise, and take the resulting boxes as the $N_e$ region proposals $\{\mathbf{R}_1, \mathbf{R}_2, \ldots, \mathbf{R}_{N_e}\}$. Here, $\mathbf{R}_\eta \in \mathbb{R}^4$ is the proposal for the $\eta$-th episode. In this way, the region proposals can be leveraged to accurately capture object from cluttered and low-quality robot observations.

**Multimodal Object Segmentation Stage.** Given a manipulation instruction such as "*Put the Eggplant in Yellow Basket*", we employ Llama 2–7B to reason over the instruction and extract the textual description of the target object to be manipulated, *e.g.*, "*Eggplant*". For the $\eta$-th episode, we then provide the target object text along with the region proposal $\{\mathbf{R}_\eta\}$ into an open-vocabulary object detector Grounding DINO (Liu et al. (2024a)) and SAM (Kirillov et al. (2023)). These models detect all relevant object instances, generate their mask annotations, and associate them with the corresponding language expressions from the target object text. We then filter the predictions based on their confidence scores, retaining only the mask annotations within the bounding box that has the highest box-confidence. Sequentially, we derive visual prompts from the object masks by randomly sampling points within the mask, generating random lines inside the object region, and extracting external bounding boxes through mask contour detection.

Finally, we apply the proposed two-stage automated annotation pipeline to the publicly available Fractal dataset (Brohan et al. (2022)) and Bridge v2 dataset (Walke et al. (2023)). We first au-

Table 1: SimplerEnv (Li et al. (2024c)) simulation evaluation results in terms of the average success rate for the Google Robot setup. VM denotes Visual Matching and VA is Variant Aggregation. ▨ and ▨ denote tuning-based methods applied to the pretrained weights of OpenVLA and $\pi_0$, respectively.

| Methods | Pick Coke Can | | Move Near | | Open/Close Drawer | | Average | |
|---|---|---|---|---|---|---|---|---|
| | VM | VA | VM | VA | VM | VA | VM | VA |
| RT-1-X (O'Neill et al. (2024)) | 56.7 | 49.0 | 31.7 | 32.3 | 59.7 | 29.4 | 49.4 | 36.9 |
| Octo-Base (Team et al. (2024)) | 17.0 | 0.6 | 4.2 | 3.1 | 22.7 | 1.1 | 14.6 | 1.6 |
| HPT (Wang et al. (2024)) | 56.0 | – | 60.0 | – | 24.0 | – | 46.7 | – |
| RoboVLMs (Liu et al. (2025)) | 72.7 | 68.3 | 66.3 | 56.0 | 26.8 | 8.5 | 56.3 | 46.3 |
| Dita (Hou et al. (2025)) | 83.7 | 85.5 | 76.0 | 73.0 | 46.3 | 37.5 | 68.7 | 65.3 |
| SpatialVLA (Qu et al. (2025)) | 81.0 | 89.5 | 69.6 | 71.7 | 59.3 | 36.2 | 71.9 | 68.8 |
| OpenVLA (Kim et al. (2024)) | 16.3 | 54.5 | 46.2 | 47.7 | 35.6 | 17.7 | 32.7 | 40.0 |
| OpenVLA-SFT | 17.5 | 51.9 | 44.6 | 42.3 | 32.8 | 16.8 | 31.6 | 38.6 |
| TraceVLA (Zheng et al. (2024)) | 28.0 | 60.0 | 53.7 | 56.4 | **57.0** | **31.0** | 46.2 | 49.1 |
| **PixelVLA** | **81.7** | **72.7** | **60.1** | **57.7** | 42.3 | 20.0 | **61.4** | **50.1** |
| $\pi_0$ (Black et al. (2024)) | 72.7 | 75.2 | 65.3 | **63.7** | 38.3 | 25.6 | 58.8 | 54.8 |
| $\pi_0$-SFT | 70.8 | 72.1 | 64.2 | 61.3 | 36.8 | 28.3 | 57.3 | 53.9 |
| **PixelVLA-$\pi_0$** | **80.7** | **76.8** | **67.7** | 62.0 | **41.3** | **30.8** | **63.3** | **56.5** |

tomatically discard samples with failed mask generation (*e.g.*, empty or invalid masks) using a simple script, and then the authors rapidly inspect the remaining samples to remove those with clearly incorrect masks. In total, this process filters out approximately 19.2% of the generated samples. The resulting dataset, Pixel-160K, contains 160K robot manipulation episodes and 6.5M image–text–action triplets with visual prompts and mask annotations.

## 4.3 VISUOMOTOR INSTRUCTION TUNING PROCEDURE

To advance fine-grained pixel-level understanding in VLAs, we propose a novel visuomotor instruction tuning procedure, consisting of a continuous action training stage and a pixel-level understanding enhancement stage. Concretely, the first continuous action training stage enables the VLA model to acquire robust continuous action representations from a large mixture of image–text–action datasets. In the second stage, pixel-level understanding is explicitly enhanced by adapting the pretrained model on Pixel-160K dataset through LoRA adaptation (Hu et al. (2022); Li et al. (2025d)). The following sections elaborate on the key designs of this two-stage training strategy.

**Continuous Action Training Stage**. Before training, we initialize the vision encoder, the MLP projector, and the LLM backbone in PixelVLA with the pretrained weights of VLAs (Kim et al. (2024); Black et al. (2024)), which has been trained on the large-scale mixture dataset OXE (O'Neill et al. (2024)). In addition, during this stage, the visual prompt-aware encoder and the multiscale pixel-aware encoder of PixelVLA are removed, while all other modules except the continuous action decoder are frozen to preserve the general manipulation knowledge learned in the pretrained VLAs. To directly map the final hidden states of the last layer of LLM to continuous action values, we follow (Zhao et al. (2023); Kim et al. (2025)) to implement L1 regression to align predicted actions generated by the proposed continuous action decoder with the ground-truth actions. Unlike Open-VLA, which represents actions as discrete tokens by normalizing each action dimension to $[-1, +1]$ and uniformly discretizing it into 256 bins, PixelVLA directly predicts continuous action values, thereby avoiding the loss of fine-grained action details introduced by discretization. Furthermore, during this stage, we train PixelVLA on a mixture of Fractal dataset and Bridge v2 dataset.

**Pixel-level Understanding Enhancement Stage**. Originally, most existing visuomotor instruction tuning methods (Li et al. (2025b); Kim et al. (2025); Yang et al. (2025)) focus on image-level understanding. In contrast, at this stage, to enhance pixel-level understanding of PixelVLA, we employ LoRA adaptation to efficiently fine-tune PixelVLA's LLM backbone on Pixel-160K dataset, while jointly training the visual prompt-aware encoder along with the multiscale pixel-aware encoder. Meanwhile, the continuous action decoder is optimized while the remaining PixelVLA modules remain frozen. Furthermore, we adopt the same L1 regression loss and continuous action representation strategy as those employed in the continuous action training stage. At each training step, given

Table 2: Evaluation results from the SimplerEnv simulation for the WidowX robot. Gra. denotes the average grasp success rate, and Suc. is the overall task completion success rate.

| Methods | Put Spoon | | Put Carrot | | Stack Blocks | | Put Eggplant | | **Average** | |
|---|---|---|---|---|---|---|---|---|---|---|
| | Gra. | Suc. | Gra. | Suc. | Gra. | Suc. | Gra. | Suc. | Gra. | Suc. |
| RT-1-X (O'Neill et al. (2024)) | 16.7 | 0.0 | 20.8 | 4.2 | 8.3 | 0.0 | 0.0 | 0.0 | 11.5 | 1.1 |
| Octo-Base (Team et al. (2024)) | 34.7 | 12.5 | 52.8 | 8.3 | 31.9 | 0.0 | 66.7 | 43.1 | 46.5 | 16.0 |
| Octo-Small (Team et al. (2024)) | 77.8 | 47.2 | 27.8 | 9.7 | 40.3 | 4.2 | 87.5 | 56.9 | 58.4 | 29.5 |
| RoboVLMs (Liu et al. (2025)) | 37.5 | 20.8 | 33.3 | 25.0 | 8.3 | 8.3 | 0.0 | 0.0 | 19.8 | 13.5 |
| SpatialVLA (Qu et al. (2025)) | 25.0 | 20.8 | 41.7 | 20.8 | 58.3 | 25.0 | 79.2 | 70.8 | 51.1 | 34.4 |
| OpenVLA (Kim et al. (2024)) | 4.1 | 0.0 | 33.3 | 0.0 | 12.5 | 0.0 | 8.3 | 4.1 | 14.6 | 1.0 |
| OpenVLA-SFT | 8.4 | 0.0 | 35.1 | 12.8 | 10.5 | 0.0 | 16.5 | 8.4 | 17.6 | 5.3 |
| **PixelVLA** | 20.8 | 4.2 | **37.5** | **20.8** | 16.6 | 0.0 | 79.2 | 41.7 | 38.5 | 16.7 |
| $\pi_0$ (Black et al. (2024)) | 45.8 | 29.1 | 25.0 | 0.0 | 50.0 | 16.6 | **91.6** | **62.5** | 53.1 | 27.1 |
| $\pi_0$-SFT | 45.3 | 26.8 | 28.6 | 4.2 | 52.3 | 18.6 | 88.5 | 59.6 | 53.7 | 27.3 |
| **PixelVLA-$\pi_0$** | **51.7** | **32.4** | 28.7 | 16.7 | **56.8** | **21.7** | 83.3 | 61.7 | **55.1** | **33.8** |

a mini-batch $\{\mathbf{x}^i, \mathbf{p}^i, \mathbf{a}^i, \mathbf{L}^i, \mathbf{V}^i\}_{i=1}^B$ sampled from the Pixel-160K dataset, the forward process at a single timestep of this stage can then be formulated as follows:

$$\mathcal{L}_{PixelVLA} = \sum_{i=1}^B \|\mathbf{a}^i - \mathcal{C}(\mathcal{H}(\mathbf{E}_v^i, \mathbf{E}_l^i, \mathbf{E}_p^i, \mathbf{E}_s^i))\|_1, \qquad (4)$$

where $\mathcal{C}(\cdot)$ refers to the continuous action decoder and $\mathcal{H}$ represents the LLM backbone of PixelVLA. In addition, $B$ denotes the mini-batch size and $\|\cdot\|_1$ is the L1 norm used for regression. Notably, $\mathbf{E}_v^i, \mathbf{E}_l^i, \mathbf{E}_o^i, \mathbf{E}_s^i$ correspond to the visual embeddings produced by the vision encoder and the MLP projector, the language embeddings from the LLM tokenizer, the pixel-aware embeddings and the prompt-aware embeddings, respectively.

## 5 EXPERIMENTS

We conduct experiments to investigate how PixelVLA leverages pixel-level understanding and multimodal prompts to enhance the performance of current VLAs in both in-domain and out-of-domain adaptation. To achieve this objective, we develop three experimental paradigms: (1) zero-shot object manipulation comparisons for out-of-domain generalization (Sec. 5.2), (2) adaptation to new robot setups to evaluate in-domain robustness (Sec. 5.3), and (3) a series of ablation studies to quantify the contribution of each individual module within PixelVLA (Sec. 5.4).

### 5.1 EXPERIMENTAL SETUP

**Evaluation tasks**. We conduct all experiments on three simulation benchmarks, *i.e.*, SimplerEnv-Google Robot (Li et al. (2024c)), SimplerEnv-WidowX (Li et al. (2024c)) and LIBERO (Liu et al. (2023a)). SimplerEnv (Li et al. (2024c)) is an open-source simulation suite that facilitates reproducible and scalable evaluation of robot manipulation policies by explicitly addressing visual and dynamic gaps between simulation and real hardware. In light of this, we conduct zero-shot object manipulation comparisons on SimplerEnv. In addition, following OpenVLA (Kim et al. (2024)), we evaluate performance of new robot adaptation across four task suites within LIBERO (Liu et al. (2023a)), *i.e.*, LIBERO-Spatial, LIBERO-Object, LIBERO-Goal and LIBERO-Long.

**Implementation Details**. To evaluate the effectiveness of PixelVLA, we apply its architecture and the proposed visuomotor instruction-tuning procedure to two widely-used VLAs, OpenVLA (Kim et al. (2024)) and $\pi_0$ (Black et al. (2024)). Regarding the training data, PixelVLA is trained in two stages: the first stage utilizes real-robot demonstrations from the Fractal dataset (Brohan et al. (2022)) and Bridge v2 dataset (Walke et al. (2023)), while the second stage employs 160K real-robot demonstrations from the proposed Pixel-160K dataset. For input robot observations across all datasets, PixelVLA is conditioned solely on a single third-person camera view and processes images at a resolution of 224×224 pixels. In all training stages, we set action chunk size to 8 for

Table 3: LIBERO Simulation Benchmark Results. We report the success rates of each method across four task suites. Models including Octo, OpenVLA, TraceVLA, Dita, SpatialVLA and PixelVLA are adapted through fine-tuning. R. represents the success rate ranking in each task suite.

| Methods | Spatial | | Object | | Goal | | Long | | Average |
|---------|---------|---------|---------|---------|---------|---------|---------|---------|---------|
| | Suc.($\uparrow$) | R.($\downarrow$) | Suc.($\uparrow$) | R.($\downarrow$) | Suc.($\uparrow$) | R.($\downarrow$) | Suc.($\uparrow$) | R.($\downarrow$) | |
| Diffusion Policy (Chi et al. (2023)) | 78.3 | 8 | 92.5 | 2 | 68.3 | 7 | 50.5 | 7 | 72.4 |
| Octo (Team et al. (2024)) | 78.9 | 7 | 85.7 | 7 | 84.6 | 4 | 51.1 | 6 | 75.1 |
| CoT-VLA (Zhao et al. (2025)) | 87.5 | 3 | 91.6 | 3 | **87.6** | 1 | 69.0 | 2 | 81.1 |
| Dita (Hou et al. (2025)) | 84.2 | 6 | **96.3** | 1 | 85.4 | 3 | 63.8 | 3 | 82.4 |
| SpatialVLA (Qu et al. (2025)) | 88.2 | 2 | 89.9 | 5 | 78.6 | 6 | 55.5 | 4 | 78.1 |
| OpenVLA (Kim et al. (2024)) | 84.7 | 4 | 88.4 | 6 | 79.2 | 5 | 53.7 | 5 | 76.5 |
| TraceVLA (Zheng et al. (2024)) | 84.6 | 5 | 89.9 | 5 | 78.6 | 6 | 55.5 | 4 | 78.1 |
| **PixelVLA** | **88.5** | 1 | 90.0 | 4 | 85.8 | 2 | **82.6** | 1 | **86.7** |

the continuous action decoder, *i.e.*, the predicted action $\mathbf{a^t} \in \mathbb{R}^{8 \times 7}$. The first training stage involves training PixelVLA for 100k steps with a batch size of 32 and a learning rate of $5 \times 10^{-4}$. Notably, in light of the effectiveness action expert in $\pi_0$, we omit the first training stage when adapting PixelVLA on $\pi_0$. During the second training stage, we fine-tune the LLM backbone of PixelVLA using LoRA adaptation with a rank $r = 32$. This stage is trained for 200k steps with a batch size of 32 and a learning rate of $1 \times 10^{-3}$. In addition, to adapt PixelVLA to the LIBERO benchmark (Liu et al. (2023a)), we fine-tune the pre-trained model for 150K steps on each task suite using LoRA adaptation with rank $r = 32$, a batch size of 32, and a learning rate of $5 \times 10^{-4}$. In addition to the two baseline VLAs, OpenVLA (Kim et al. (2024)) and $\pi_0$ (Black et al. (2024)), we compare the performance of PixelVLA against other state-of-the-art VLAs, such as RT-1 (Brohan et al. (2022)), HPT (Wang et al. (2024)), Octo (Team et al. (2024)), TraceVLA (Zheng et al. (2024)), RoboVLMs (Liu et al. (2025)), Dita (Hou et al. (2025)) and SpatialVLA (Qu et al. (2025)). To further ensure fairness, we additionally fine-tune $\pi_0$ and OpenVLA on the Fractal and Bridge datasets, obtaining baselines denoted as $\pi_0$-SFT and OpenVLA-SFT.

## 5.2 ZERO-SHOT OBJECT MANIPULATION COMPARISONS

This subsection evaluates the zero-shot manipulation performance of our model against baseline VLAs across multiple task categories and robot platforms. As shown in Tab. 1, on the Google Robot setup PixelVLA achieves an average VM score of 61.4 and VA score of 50.1, surpassing OpenVLA by 28.7/10.1 and OpenVLA-SFT by 29.8/11.5 in VM/VA, respectively. These results indicate a strong capability in both pixel-level understanding and adaptation to textual and visual prompts in out-of-domain adaptation. Notably, as shown in Fig. 4, PixelVLA outperforms TraceVLA and OpenVLA across various environmental variations, highlighting the effectiveness of the proposed visuomotor instruction tuning procedure in addressing out-of-domain generalization.

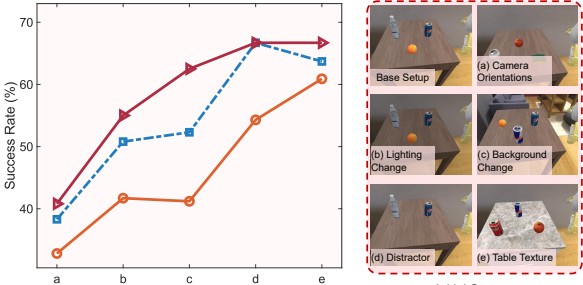

Figure 4: Performance comparison of OpenVLA, TraceVLA and PixelVLA performance across various environmental variations on SimplerEnv-Google Robot setup: camera orientations, lighting, background, distractors, and table texture.

Similar trends are observed on the WidowX robot setup in Tab 2, where PixelVLA-$\pi_0$ achieves an average grasp score of 55.1 and success score of 16.7, outperforming the baseline $\pi_0$ by 2.0 and 6.7, respectively, and surpassing RoboVLM by 35.3 and 18.2. The results strongly affirm that PixelVLA's architectural innovations including its multiscale pixel-aware encoder and integration of visual prompts, significantly enhance its zero-shot perceptual and operational capabilities. Furthermore, the significant improvements of PixelVLA and PixelVLA-$\pi_0$ over the baselines OpenVLA

and $\pi_0$ demonstrate that incorporating the finer-grained pixel-level spatial comprehension into existing VLAs enables more effective adaptation to unseen objects.

## 5.3 NEW ROBOT SETUPS ADAPTATION COMPARISONS

To evaluate the adaptability of PixelVLA to novel robotic setups and task configurations, we employ the proposed automated annotation pipeline to process the LIBERO benchmark training data (Liu et al. (2023a)), yielding the LIBERO-Pixel dataset. Subsequently, as summarized in Tab. 3, PixelVLA achieves state-of-the-art performance with an average success rate of 86.7 across all tasks, significantly surpassing strong baselines. In addition, As shown in Tab. 3, PixelVLA outperforms OpenVLA across all four LIBERO task suites, with particularly large gains on LIBERO-Long. We attribute this to the Continuous Action Decoder: action chunking helps the policy capture longer-range temporal dependencies (Liu et al. (2024b)), while continuous action prediction mitigates compounding discretization errors (Black et al. (2024)) over long-horizon manipulation. These superior results demonstrate the enhanced adaptability of PixelVLA to new robotic setups, highlighting the effectiveness of pixel-level visual understanding and continuous action representation in PixelVLA. Notably, PixelVLA achieves significant performance in the LIBERO-Long setup, demonstrating its effectiveness in long-range manipulation.

## 5.4 ABLATION STUDIES

This subsection evaluates the effectiveness of individual components in PixelVLA on SimplerEnv-Google Robot in terms of Variant Aggregation. As shown in Tab. 4, we use OpenVLA as the baseline model. Here, Baseline+FT refers to fine-tuning OpenVLA directly on a mixture of Fractal dataset and Bridge v2 dataset. In addition, Baseline+FT+CAT indicates training OpenVLA with the proposed continuous action training stage using a continuous action decoder, while Baseline+FT+PUE denotes fine-tuning OpenVLA with the proposed pixel-level understanding enhancement stage on the Pixel-160 dataset.

Table 4: Quantitative ablation studies on Variant Aggregation for the Google Robot setup, evaluated in the SimplerEnv simulation environment (Li et al. (2024c)).

| Methods | Pick Coke Can | Move Near | Open/Close Drawer | Average |
|---|---|---|---|---|
| Baseline | 54.5 | 47.7 | 17.7 | 40.0 |
| +FT | 51.9 | 42.3 | 16.8 | 37.0 |
| +FT+CAT | 61.3 | 52.3 | 17.7 | 43.8 |
| +FT+PUE | 71.1 | 54.7 | **21.3** | 48.0 |
| **PixelVLA** | **72.7** | **57.7** | 20.0 | **50.1** |

As presented in Tab. 4, incorporating the continuous action training stage (Baseline+FT+CAT) improves the average score of 3.8% compared to Baseline, highlighting the benefits of the proposed continuous action decoder. Further enhancement with pixel-level understanding (Baseline+FT+PUE) yields a more substantial gain of 8.0%. Compared to the single-stage Baseline+FT+PUE, PixelVLA adds a continuous action training stage, and this two-stage scheme leads to a slight drop on Open/Close Drawer, due to the catastrophic forgetting (Li et al. (2024a)) in joint two-stage optimization and the high difficulty and sensitivity of this task. Ultimately, PixelVLA outperforms Baseline+FT+CAT by 6.3%. This progressive improvement validates the effectiveness of both pixel-level understanding and multimodal prompts in advancing generalization capabilities.

## 6 CONCLUSION

This paper proposes PixelVLA, a vision-language-action (VLA) model, to address the limitations of existing VLAs, such as insufficient pixel-level understanding and over-reliance on textual prompts. PixelVLA integrates a multiscale pixel-aware encoder to inject pixel-level understanding, a continuous action decoder for generating accurate robotic actions, and a lightweight visual prompt-aware encoder to support both textual and visual prompts. In addition, a two-stage automated annotation pipeline is designed to construct the Pixel-160K dataset containing 160K manipulation episodes. To advance fine-grained pixel-level understanding in VLAs, we propose a novel two-stage visuomotor instruction tuning framework to train PixelVLA, requiring only 1.5% of the pretraining cost of OpenVLA. Expensive evaluations on three VLA benchmarks show that PixelVLA can be integrated into existing VLAs to achieve a $10.1\% \sim 28.7\%$ improvement in manipulation success rate, effectively enhancing the spatial comprehension and complex environment adaptability of VLAs.

ACKNOWLEDGMENTS

This work is supported by the National Key Research and Development Program of China (2023YFB4704800) and National Nature Science Foundation of China under Grant (62225310,62273333), and the Fundamental Research Funds for the Central Universities (2025ZYGXZR032, 2024ZYGXZR024).

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

# A APPENDIX

This supplemental material introduces additional details not mentioned in the main paper. Overall, it mainly includes the following aspects:

**(1)** Details of Pixel-160K dataset in Sec. A.1.

**(2)** More experiment setup in Sec. A.2.

**(3)** More implementation details in Sec. A.3.

**(4)** More qualitative comaprisons in Sec. A.4.

**(5)** Limitations in Sec. A.5.

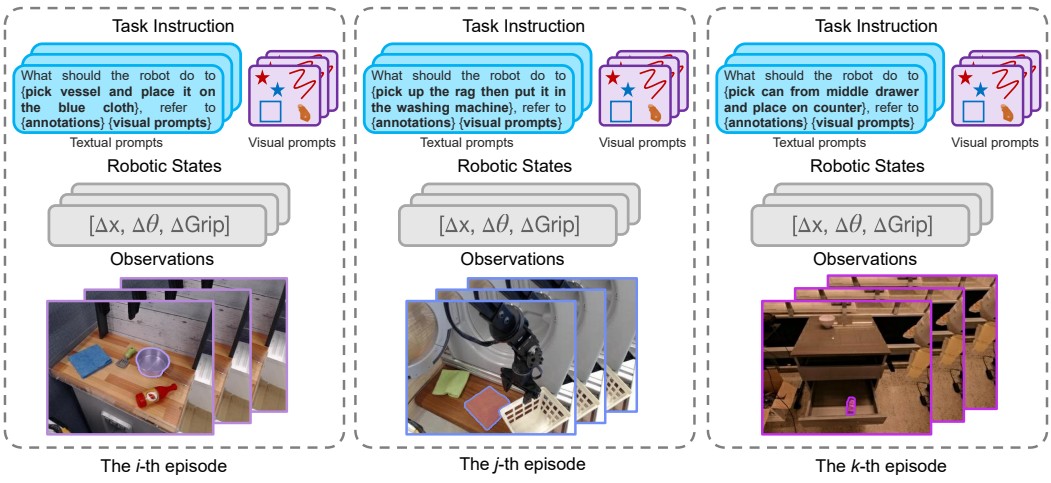

Figure 5: The episode example in our Pixel-160K dataset.

## A.1 PIXEL-160K DATASET

This section provides a detailed description of the Pixel-160K dataset and the proposed automated annotation pipeline. In contrast to existing robot datasets composed of image-level and text instructions, Pixel-160K offers fine-grained pixel-level annotations and supports both textual and visual prompts, aiming to train VLAs for more precise pixel-level spatial understanding and diversified human–robot interaction. Pixel-160K comprises 160K robot manipulation episodes and 6.5M image–text–action triplets enriched with visual prompts and mask annotations. Specifically, for the $i$-th episode $\mathbf{E}_i$ in Pixel-160K dataset, $\mathbf{E}_i = \{\mathbf{X}_i, \mathbf{A}_i, \mathbf{P}_i, \mathbf{L}_i, \mathbf{V}_i\}$, where $\mathbf{X}_i = \{\mathbf{x}_i^t\}_{t=1}^T$, $\mathbf{P}_i = \{\mathbf{p}_i^t\}_{t=1}^T$, $\mathbf{A}_i = \{\mathbf{a}_i^t\}_{t=1}^T$. As shown in Fig. 5, to effectively inject the pixel-level annotations and visual masks, we reformulate the textual instruction–*What should the robot do to* {*</instruction>*} as *What should the robot do to* {*</instruction>*}, *refer to* {*</annotations>*} {*</visual prompts>*}. Here, *</·>* a placeholder and is subsequently replaced with the language embeddings, the pixel-aware embeddings and the prompt-aware embeddings.

As illustrated in Fig. 6(a), the proposed automated annotation pipeline consists of two sequential stages: (a) Gripper-aware Region Proposal Stage and (b) Multimodal Object Segmentation Stage. The gripper-aware region proposal stage consists of four steps:

**1. Gripper-close State Extraction**: For each manipulation episode, we parse the robotic state sequence $\Delta x, \Delta\theta, \Delta\text{Grip}$. Frames in which the gripper state $\Delta\text{Grip} = 1$ are identified as gripper-close states, indicating that the gripper is interacting with an object. The first frame in each episode is selected as the key observation $\mathbf{x}_\eta^{G_\eta}$, representing the earliest moment of potential object contact.

**2. Discrete Video Composition**: We collect the gripper-close observations from all episodes, *i.e.*, $\{\mathbf{x}_1^{G_1}, \mathbf{x}_2^{G_2}, \ldots, \mathbf{x}_{N_e}^{G_{N_e}}\}$, where $N_e$ is the total number of episodes. These frames are then organized

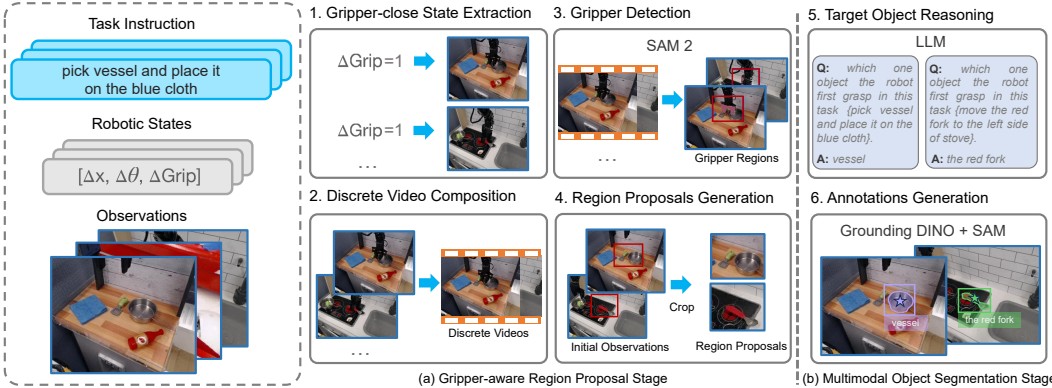

Figure 6: The proposed automated annotation pipeline for generating visual prompts and mask annotations at scale for a given robot dataset, consisting of a gripper-aware region proposal stage and a multimodal object segmentation stage.

into a discrete video set that captures diverse gripper-object interaction states across the dataset, providing compact yet informative cues for region localization.

**3. Gripper Detection**: To precisely localize the gripper within the selected frames, we apply SAM 2 (Ravi et al. (2024)). The model segments the gripper region from each observation, producing bounding boxes tightly enclosing the gripper. These gripper regions serve as reliable anchors, as the manipulated objects typically appear adjacent to or in contact with the gripper.

**4. Region Proposals Generation**: Finally, we crop the localized gripper regions from the initial observations to form region proposals $\mathbf{R}_1, \mathbf{R}_2, \ldots, \mathbf{R}_{N_e}$, with each $\mathbf{R}_\eta \in \mathbb{R}^4$ denoting a bounding box for the $\eta$-th episode. These proposals effectively filter out irrelevant background clutter and highlight the object-relevant areas, thereby facilitating accurate segmentation in the subsequent multimodal object segmentation stage.

As illustrated in Fig. 6(b), the multimodal object segmentation stage consists of two steps:

**5. Target Object Reasoning**: Given a manipulation instruction, such as "Pick the vessel and place it on the blue cloth" or "Move the red fork to the left side of the stove", we employ a large language model (LLM) to parse the instruction and extract the target object to be manipulated. The LLM identifies the first object the robot needs to grasp and outputs its textual description (*e.g.*, "vessel" or "the red fork"). This reasoning step transforms natural-language task instructions into precise object queries, enabling alignment between language and perception.

**6. Annotations Generation**: For the $\eta$-th episode, we feed the target object text together with the region proposal $\mathbf{R}_\eta$ into an open-vocabulary detection and segmentation pipeline comprising Grounding DINO (Liu et al. (2024a)) and SAM (Kirillov et al. (2023)). Grounding DINO detects candidate object bounding boxes conditioned on the target object text, grounding the language query into specific image regions. Subsequently, SAM refines these detections by generating pixel-level segmentation masks for the grounded boxes. The outputs include bounding boxes, segmentation masks, and text-object alignments. We then filter the results by selecting the mask with the highest confidence score within the region proposal, discarding low-confidence or irrelevant detections. Finally, from the retained object mask, we derive visual prompts by: randomly sampling points inside the mask, generating random lines within the object area, and extracting external bounding boxes via contour detection.

A.2    EXPERIMENT SETUP

**Evaluation tasks**. As shown in Fig. 7, we conduct all experiments on three simulation benchmarks, *i.e.*, SimplerEnv-Google Robot (Li et al. (2024c)), SimplerEnv-WidowX (Li et al. (2024c)) and LIBERO (Liu et al. (2023a)). Specifically, SimplerEnv Li et al. (2024c) is an open-source simulation suite designed for reproducible and scalable evaluation of robot manipulation policies. It explicitly

Figure 7: Experimental Setup Overview. We evaluate PixelVLA on three simulation benchmarks: SimpleEnv-WidowX with WidowX robot, SimpleEnv-Google Robot with Google robot, and LIBERO with Franka.

addresses the visual and dynamic gaps between simulation and real hardware, enabling more faithful assessments of policy generalization. SimplerEnv provides two complementary evaluation settings: Visual Matching (VM), which minimizes visual appearance discrepancies to improve the correlation between simulation and real-world performance; and Variant Aggregation (VA), which, inspired by domain randomization, introduces diverse visual perturbations and aggregates results across multiple randomized environments to obtain more robust performance estimates. Building on these capabilities, we perform zero-shot object manipulation experiments on SimplerEnv to benchmark the effectiveness of our approach.

As shown in Fig. 7, LIBERO (Liu et al. (2023a)) is a comprehensive benchmark suite designed to evaluate continual adaptation and generalization in robot manipulation. It consists of multiple task suites that focus on different aspects of knowledge transfer: LIBERO-Spatial, which requires learning new spatial relationships between identical objects; LIBERO-Object, which involves recognizing and manipulating novel object types; and LIBERO-Goal, which evaluates the ability to adapt to new task goals given the same objects and spatial layouts. Additionally, LIBERO-Long extends the challenge to long-horizon tasks, testing a robot's capacity for compositional reasoning and extended action planning. Together, these task suites provide a systematic framework for studying declarative knowledge (spatial and object-level) and knowledge (goal-oriented behaviors) in robot learning.

## A.3 IMPLEMENTATION DETAILS

**Training**. In the first stage of continuous action training, we initialize PixelVLA's vision encoder, MLP projector, and LLM backbone with pretrained weights from VLAs (Kim et al. (2024); Black et al. (2024)), which were trained on the large-scale OXE dataset (O'Neill et al. (2024)). To focus on learning continuous action mappings, the visual prompt-aware encoder and the multiscale pixel-aware encoder are removed, while all other modules except the continuous action decoder are frozen to preserve general manipulation knowledge. The continuous action decoder maps the final hidden states of the LLM to continuous action values. We use L1 regression (Zhao et al. (2023); Kim et al. (2025)) to align the predicted actions with ground-truth actions. Each continuous action dimension is uniformly discretized into 256 bins and normalized to $[-1, +1]$ for fine-grained prediction. The input observation consists of a single third-person camera view resized to $224 \times 224$ pixels. Actions are predicted in chunks of 8 timesteps, with $\mathbf{a^t} \in \mathbb{R}^{8 \times 7}$. During this stage, PixelVLA is trained on a mixture of Fractal (Brohan et al. (2022)) and Bridge v2 (Walke et al. (2023)) datasets. Training runs for 100k steps with a batch size of 32 and a learning rate of $5 \times 10^{-4}$. This stage is omitted when adapting PixelVLA to $\pi_0$ due to the effectiveness of its pretrained action expert.

In Pixel-level Understanding Enhancement Stage, PixelVLA is fine-tuned for pixel-level understanding using the Pixel-160K dataset. The visual prompt-aware encoder and multiscale pixel-aware encoder are jointly trained, while LoRA adaptation is applied to the LLM backbone (rank $r = 32$). All other modules remain frozen except the continuous action decoder, which is optimized using L1 regression, following the same continuous action representation as in the first stage. Input observations are a single third-person camera view resized to $224 \times 224$ pixels. Actions are predicted in chunks of 8 timesteps, $\mathbf{a^t} \in \mathbb{R}^{8 \times 7}$. Training runs for 200k steps with a batch size of 32 and a

Table 5: SimplerEnv (Li et al. (2024c)) simulation evaluation results in terms of the average success rate for the Google Robot setup. VM denotes Visual Matching and VA is Variant Aggregation. ▣ denotes tuning-based methods applied to the pretrained weights of OpenVLA.

| | Methods | Pick Coke Can | | | | Move Near | Open/Close Drawer | | | Average |
| | | Horizontal | Vertical | Standing | **Average** | **Average** | Open | Close | **Average** | |
|---|---|---|---|---|---|---|---|---|---|---|
| **VM** | RT-1-X (O'Neill et al. (2024)) | 82.0 | 33.0 | 55.0 | 56.7 | 31.7 | 29.6 | 89.1 | 59.7 | 53.4 |
| | Octo-Base (Team et al. (2024)) | 21.0 | 21.0 | 9.0 | 17.0 | 4.2 | 0.9 | 44.4 | 22.7 | 16.8 |
| | HPT (Wang et al. (2024)) | – | – | – | 56.0 | 60.0 | – | – | 24.0 | 46.0 |
| | RoboVLMs (Liu et al. (2025)) | 85.0 | 43.0 | 90.0 | 72.7 | 66.3 | 28.7 | 25.0 | 26.8 | 56.3 |
| | SpatialVLA (Qu et al. (2025)) | 70.0 | 82.0 | 91.0 | 81.0 | 69.6 | 49.1 | 69.4 | 59.3 | 71.9 |
| | OpenVLA (Kim et al. (2024)) | 27.0 | 3.0 | 19.0 | 16.3 | 46.2 | 19.4 | 51.8 | 35.6 | 27.7 |
| | TraceVLA (Zheng et al. (2024)) | – | – | – | 28.0 | 53.7 | – | – | 57.0 | 42.0 |
| | **PixelVLA** | 90.9 | 63.6 | 90.9 | 81.7 | 60.1 | 25.3 | 59.3 | 42.3 | 65.0 |
| **VA** | RT-1-X (O'Neill et al. (2024)) | 56.9 | 20.4 | 69.8 | 49.0 | 32.3 | 6.9 | 51.9 | 29.4 | 39.6 |
| | Octo-Base (Team et al. (2024)) | 0.5 | 0.0 | 1.3 | 0.6 | 3.1 | 0.0 | 2.1 | 1.1 | 1.1 |
| | HPT (Wang et al. (2024)) | – | – | 60.0 | – | – | – | – | – | – |
| | RoboVLMs (Liu et al. (2025)) | 77.8 | 48.0 | 79.1 | 68.3 | 56.0 | 1.6 | 15.3 | 8.5 | 46.3 |
| | SpatialVLA (Qu et al. (2025)) | 93.3 | 83.1 | 92.0 | 89.5 | 71.7 | 23.3 | 49.2 | 36.2 | 68.8 |
| | OpenVLA (Kim et al. (2024)) | 71.1 | 27.1 | 65.3 | 54.5 | 47.7 | 15.8 | 19.5 | 17.7 | 39.8 |
| | TraceVLA (Zheng et al. (2024)) | – | – | – | 60.0 | 56.4 | – | – | 31.0 | 45.0 |
| | **PixelVLA** | 81.8 | 54.5 | 81.8 | 72.7 | 57.7 | 22.8 | 16.4 | 20.0 | 52.6 |

learning rate of $1 \times 10^{-3}$, enhancing PixelVLA's ability to localize and manipulate target objects at the pixel level.

**Inference**. During inference, the user only needs to provide a semantic instruction along with visual prompts in the initial observation. Subsequently, we incorporate FastSAM to predict the mask annotation of the target object based on the given visual prompts. This introduces an additional computational overhead of approximately 1% of the overall model inference cost, resulting in a negligible impact on inference efficiency.

## A.4 QUALITATIVE COMPARISONS

As shown in Tab. 5, this section presents the evaluation results of the simpler environment on the Google Robotics benchmark, which includes tasks such as Coke can manipulation (horizontal, vertical and standing picking) and drawer operations (open and close). PixelVLA demonstrates exceptional performance, achieving an average VM score of 65.0, significantly outperforming OpenVLA (27.7) and TraceVLA (42.0). In the VA category, PixelVLA achieves an average score of 52.6, again surpassing OpenVLA (39.8) and TraceVLA (45.0). These results indicate that PixelVLA outperforms its counterparts in both pixel-level understanding and the adaptation to visual prompts, showcasing a remarkable capacity for generalist manipulation across the evaluated tasks. The significant margins in performance metrics underscore the effectiveness of the proposed method in addressing out-of-domain generalization challenges.

## A.5 LIMITATIONS

Although PixelVLA substantially improves VLA performance through pixel-level understanding and tailored visual prompts, it remains limited in handling richer input modalities (e.g., 3D perception) and more advanced forms of visual prompting, such as precise trajectory guidance, reference-image prompts, pose-conditioned prompts, or compositional prompt sequences. In addition, while PixelVLA is extensively validated across three simulated benchmarks, we expect that real-world robot experiments would further strengthen its contributions. We regard these extensions as important directions for future work.

