# OpenReview forum: "PixelVLA: Advancing Pixel-level Understanding in Vision-Language-Action Model"
_ICLR.cc/2026/Conference — ICLR 2026 Poster_

### Official Review · Reviewer_FnpE · 2025-10-22

**Soundness:** 2
**Presentation:** 2
**Contribution:** 3
**Rating:** 4
**Confidence:** 4

**Summary:**

The authors introduce a pixel encoder to condition VLAs on visual prompts, construct a suitable automated data annotation pipeline, and instruction tune their modified VLA to gain pixel level awareness. Their proposed visual prompt encoder conditions on multi-scale features of the robot visual observation and its resulting tokens are consumed by an LLM along with regular visual and textual tokens. The LLM outputs are mapped to robot actions with a suitable action decoder.
They integrate their proposed PixelVLA into two strong VLAs, OpenVLA and Pi0. Extensive experiments highlight performance improvements from their framework.

**Strengths:**

1) The authors conduct extensive experiments to highlight improvements over SOTA baselines
2) Simple idea of incorporating pixel prompts
3) Clever multi-stage training setup to benefit from existing VLA's pretrained weights
4) Thorough ablations

**Weaknesses:**

**1) Architecture is unclear**
  - In Figure 2, part (a) "Multiscale Pixel-aware Encoder" also shows another Visual Prompting Decoder which is unclear. This figure needs to be explained better and updated suitably.
  - "Visual Prompting Decoder" is not explained clearly. What is its exact input structure? How is does it "preserve the spatial positional information" of visual prompts? These need to be explained.
  - How exactly are the visual prompts represented? Fig 2 shows them both overlaid on image and input separately. The separate input needs to be described more clearly.

**2) Unclear Data Generation**
  - "detect the gripper to generate N_e region proposals" - how are region proposals generated from gripper location?
  - L262 "we employ a LLM" - which one? How is it prompted?
  - "we filter out approximately 19.2% failed samples" - how are they filtered? Manually by humans?

**3) Unclear Training Details**
  - L307 "uniformly discretize each continuous action representation into 256 bins" - why? PixelVLA uses a continuous action decoder? The reasoning behind this is highly unclear.

**4) Related Work**
  - Consider discussing related VLM work Ferret (https://arxiv.org/pdf/2310.07704) and FerretV2 that performs similar visual prompt encoding to improve VLA performance.

**Questions:**

See weaknesses. The paper is interesting and contains strong experimental evidence. However, several missing key details weakens it in current form.

---

> ### Author Response · Authors · 2025-11-22
> **Official Comment by Authors**
>
> We first thank the reviewer for recognizing our contributions, including the extensive experiments over strong SOTA baselines, the simple yet effective use of pixel prompts, the multi-stage training strategy that leverages pretrained VLA weights, and the thorough ablations. We also appreciate the reviewer’s positive assessment of the paper’s overall interest and experimental strength.
>
> > **W1:** Architecture is unclear
>
> **A1:** We apologize for the confusion and thank the reviewer for pointing out these issues. To address them, we have redrawn Fig. 2 and expanded the description of the Visual Prompting Decoder in the revised version. Here is a detailed response to the three questions below:
>
> > **W1.1:** Unclear depiction of visual prompting decoder in Fig. 2(a)
>
> **A1.1:** To avoid ambiguity, we now explicitly distinguish the two branches in Fig. 2:
>
> 1) On the left side, what was previously referred to as a “Visual Prompting Decoder” is now renamed to Visual Prompt-aware Encoder, highlighting that it operates on visual prompts at the encoding stage.
>
> 2) On the right side, we separate and depict the pipelines of (a) Multiscale Pixel-aware Encoder and (b) Visual Prompt-aware Encoder to clearly show how each branch processes its inputs and how their outputs are fused.
>
> We have updated this caption throughout the revised manuscript.
>
> > **W1.2:** Details missing for visual prompting decoder and its spatial positional information preservation
>
> **A1.2:** As shown in Fig. 2(b), we adopt a lightweight prompt encoder similar to that in SAM and integrate it into PixelVLA as the visual prompting encoder. Concretely, the user-provided prompts $\mathbf{V}^0 \in \mathbb{R}^{H \times W}$ are first converted into continuous positional embeddings based on their normalized image coordinates, and then combined with learned prompt-type embeddings to produce prompt features $\mathbf{F}^0_{s} \in \mathbb{R}^{N_{s} \times D_s}$, where $N_s$ is the embedding length and $D_s$ is the feature dimension. These features $\mathbf{F}^0_{s}$ are further transformed by an MLP to obtain the final prompt-aware embeddings $\mathbf{E}^0_s \in \mathbb{R}^{N_s \times D}$. Since each embedding is explicitly tied to a specific location or region in the image via its coordinate-based positional embedding, the spatial positional information of the visual prompts is preserved throughout the encoding process. We have incorporated these additional details about the Visual Prompt-aware Encoder into Section 4.1 PixelVLA Architecture of the revised manuscript.
>
> > **W1.3:** Clarifying the separate input representation of visual prompts
>
> **A1.3:** We agree that the previous depiction of visual prompts was confusing. In the revised Fig. 2, we explicitly show visual prompts as a separate input branch feeding into the Visual Prompt-aware Encoder. These clarifications and the updated figure are included in the revised manuscript to make the architecture and the role of visual prompts much clearer.
>
> > **W2:** Unclear Data Generation
>
> **A2:** We thank the reviewer for pointing out these missing implementation details. We have added the descriptions below to the revised version.
>
> > **W2.1:** How are region proposals generated from the gripper location?
>
> **A2.1:** Concretely, after obtaining the gripper masks from SAM2, we first compute the minimal axis-aligned bounding boxes that tightly enclose the masks. We then uniformly enlarge this bounding boxes by a fixed margin on all sides, both to capture local contextual information and to mitigate minor detection noise, and use the resulting boxes as the final region proposals.
>
> > **W2.2:** Which LLM is used and how is it prompted?
>
> **A2.2:** In our data generation pipeline, we use Llama 2–7B as the LLM. As shown in Step 5 of Fig. 6 in the supplementary material, the LLM is prompted with: “Which object does the robot grasp first in this task: {Task Instruction}?”
>
> > **W2.3:** How are failed samples filtered?
>
> **A2.3:** Failed samples are filtered in two stages. We first automatically discard samples with failed mask generation (e.g., empty or invalid masks) using a simple script, and then the authors rapidly inspect the remaining samples to remove those with clearly incorrect masks. In total, this process filters out approximately 19.2\% of the generated samples.

---

> > ### Comment · Reviewer_FnpE · 2025-11-24
> > **Final Comments**
> >
> > The paper looks great in the current format. The rebuttal addressed most concerns raised.
> >
> > Not sure if inference costs (runtime) of method is reported in the paper (or if it changes compared to baselines). Consider adding this in the paper too. This will be beneficial.

---

> > > ### Author Response · Authors · 2025-11-27
> > > **Official Comment by Authors**
> > >
> > > Thank you again for your time and for these very helpful suggestions that have substantially improved the paper. We are glad to hear that our previous responses addressed most of your concerns.
> > >
> > > > **W5:** Additional visual prompting works for strengthening motivation
> > >
> > > **A5:** Thanks for your advice.
> > >
> > > 1) Following your advice, in the revised Introduction, we highlight how visual prompting has been shown to improve robustness an spatial understanding in VLMs to benefit our motivation. The sentence has been revised to:
> > >
> > > “*The second limitation lies in prompting. Most VLAs depend solely on textual instructions, which overlook subtle visual cues and constrain spatial awareness **[11]** and multimodal human–robot interaction [12-13].*”
> > >
> > > 2) We have also incorporated the two additional references into the Related Work discussion, emphasizing how their findings on improved spatial reasoning and referential understanding further support the motivation behind PixelVLA. These revisions are summarized as follows:
> > >
> > > “***Visual prompting in VLMs.** Visual prompting methods [1-4] have recently emerged as a complementary paradigm to textual prompting, allowing models to accept more fine-grained supervision in the form of region-level [5] and even pixel-level instructions [6-7] over multimodal inputs. Shikra **[8]** extends MLLMs with a simple vision–encoder–LLM architecture that treats spatial coordinates as natural language tokens. Ferret [9] enhances region-level grounding in MLLMs through a hybrid region representation and a spatial-aware visual sampler that supports diverse region inputs, while Ferret-v2 [10] further introduces any-resolution grounding and multi-granularity visual encoding, leading to improved fine-grained visual understanding and localization over prior MLLMs. LocVLM **[11]** proposes an image-space coordinate–based instruction fine-tuning framework for V-LLMs that explores coordinate representations and spatial objectives to inject spatial awareness. However, despite these advances, robust pixel-level understanding in VLA frameworks remains challenging, especially when aligning fine-grained spatial cues with continuous, high-precision action control.*”
> > >
> > > [1] Vpgtrans: Transfer visual prompt generator across llms.
> > >
> > > [2] Pixellm: Pixel reasoning with large multimodal model.
> > >
> > > [3] Visionllm v2: An end-to-end generalist multimodal large language model for hundreds of vision-language tasks.
> > >
> > > [4] Omg-llava: Bridging image-level, object-level, pixel-level reasoning and understanding.
> > >
> > > [5] Regiongpt: Towards region understanding vision language model.
> > >
> > > [6] Groma: Localized visual tokenization for grounding multimodal large language models.
> > >
> > > [7] Glamm: Pixel grounding large multimodal model.
> > >
> > > **[8] Shikra: Unleashing multimodal llm's referential dialogue magic.**
> > >
> > > [9] Ferret: Refer and Ground Anything Anywhere at Any Granularity.
> > >
> > > [10] Ferret-v2: An improved baseline for referring and grounding with large language models.
> > >
> > > **[11] Learning to localize objects improves spatial reasoning in visual-llms.**
> > >
> > > [12] Vima: Robot manipulation with multimodal prompts]
> > >
> > > [13] Tracevla: Visual trace prompting enhances spatial-temporal awareness for generalist robotic policies
> > >
> > >
> > > > **W6:** Suggestion to report inference cost compared to baselines.
> > >
> > > **A6:** Thank you very much for this constructive suggestion. We report the inference efficiency comparison in Tab.6 in our supplemental material. PixelVLA increases OpenVLA’s throughput from 6.2 Hz to 9.5 Hz and reduces per-step latency from 161.3 ms to 105.3 ms, showing that the continuous action decoder brings substantial inference gains by replacing token-wise autoregressive decoding with single-shot continuous action prediction. Compared to $\pi_0$, PixelVLA-$\pi_0$ introduces only a minor overhead (11.6→11.0 Hz, 86.5→91.3 ms), which is negligible in most applications, while yielding significantly better manipulation performance in our experiments.
> > >
> > > The inference efficiency comparison results are summarized in Table 6 below:
> > >
> > > | Methods      | Params (↓) | Throughput (Hz) (↑) | Infer. Latency (ms) (↓) |
> > > |-------------|-----------:|--------------------:|------------------------:|
> > > | OpenVLA     | 7B         | 6.2                 | 161.3                   |
> > > | **PixelVLA**    | 7B         | 9.5                 | 105.3                   |
> > > | π₀          | 3B         | 11.6                | 86.5                    |
> > > | **PixelVLA-π₀** | 3B         | 11.0                | 91.3                    |
> > >
> > > We have added these inference efficiency results and the above analysis to the supplementary material of the revised version.

---

> ### Author Response · Authors · 2025-11-22
> **Official Comment by Authors**
>
> > **W3:** Unclear Training Details
>
> **A3:** We thank the reviewer for catching our mistake at L307. The original wording was misleading and we have corrected it in the revised version.
> Specifically, we now clarify that this discretization only applies to the OpenVLA baseline, not to PixelVLA. The sentence has been revised to:
>
> “*Unlike OpenVLA, which represents actions as discrete tokens by normalizing each action dimension to [−1,+1] and uniformly discretizing it into 256 bins, PixelVLA directly predicts continuous action values, thereby avoiding the loss of fine-grained action details introduced by discretization.*”
>
> > **W4:** Consider discussing related VLM works that performs similar visual prompt encoding.
>
> **A4:** We thank the reviewer for this helpful suggestion. We have expanded the Related Work section with a dedicated paragraph on visual prompting in VLMs, explicitly connecting prior visual-prompt encoding methods to our design in PixelVLA. In particular, we added the following discussion:
>
>
> “***Visual prompting in VLMs.** Visual prompting methods [1-4] have recently emerged as a complementary paradigm to textual prompting, allowing models to accept more fine-grained supervision in the form of region-level and even pixel-level instructions [5-6] over multimodal inputs. RegionGPT [7] improves region-level understanding in VLMs by enhancing the spatial awareness of visual encoders. Ferret [8] enhances region-level grounding in MLLMs through a hybrid region representation and a spatial-aware visual sampler that supports diverse region inputs, while Ferret-v2 [9] further introduces any-resolution grounding and multi-granularity visual encoding, leading to improved fine-grained visual understanding and localization over prior MLLMs. However, despite these advances, robust pixel-level understanding in VLA frameworks remains challenging, especially when aligning fine-grained spatial cues with continuous, high-precision action control.*”
>
> [1] Vpgtrans: Transfer visual prompt generator across llms.
>
> [2] Pixellm: Pixel reasoning with large multimodal model.
>
> [3] Visionllm v2: An end-to-end generalist multimodal large language model for hundreds of vision-language tasks.
>
> [4] Omg-llava: Bridging image-level, object-level, pixel-level reasoning and understanding.
>
> [5] Groma: Localized visual tokenization for grounding multimodal large language models.
>
> [6] Glamm: Pixel grounding large multimodal model.
>
> [7] Regiongpt: Towards region understanding vision language model.
>
> [8] Ferret: Refer and Ground Anything Anywhere at Any Granularity.
>
> [9] Ferret-v2: An improved baseline for referring and grounding with large language models.

---

> > ### Comment · Reviewer_FnpE · 2025-11-24
> > **More on "Visual prompting in VLMs"**
> >
> > This is great. Maybe consider using this to draw motivation for your method too, i.e. how such visual prompting worked great in the image domain.
> >
> > Also noting two more related works:
> >   - Learning to Localize Objects Improves Spatial Reasoning in Visual-LLMs
> >   - Shikra: Unleashing Multimodal LLM's Referential Dialogue Magic
> > These particularly show how training with visual prompting method improve the general robustness and spatial understanding of the underlying models, which will directly benefit your motivation.

---

### Official Review · Reviewer_DBFX · 2025-11-01

**Soundness:** 3
**Presentation:** 3
**Contribution:** 2
**Rating:** 6
**Confidence:** 4

**Summary:**

The authors present PixelVLA, a VLA model that introduces diverse multimodal prompts to the VLA pipeline and whose architecture is designed and trained specifically for these prompts. The VLA itself consists of a vision encoder to process image observations, visual prompting encoder (from SAM) and multi-scale visual encoder to process multimodal prompts (points, boxes, lines etc), and LLM backbone and a continuous action decoder. To generate visuomotor instruction tuning data for training PixelVLA, the authors use an LLM to extract information about target objects, and use an object detector and segmentation model to extract multimodal prompts, all over the Fractal and Bridge datasets, and contribute the resulting dataset. The experiments are done over SimplerEnv and Libero, across multiple VLA backbones, to show the benefits of PixeVLA.

**Strengths:**

- Meaningful technical contribution in Pixel-160k dataset and in constructing a VLA that takes advantage of multimodal prompts.
- Experiments test two SOTA VLA architectures, showing that PixelVLA can be built on top of multiple types of VLAs.

**Weaknesses:**

- Unclear whether method is feasible to transfer to novel environments, due to lack of real world experiments.
- Analysis of results are lacking and leave out failure cases; for example, why does PixelVLA perform so well on Libero Long but struggle on the Object/Goal splits (Table 3)? Why would the pixel-level understanding training damage performance on the open/close drawer task (Table 4)?

**Questions:**

l. 218 Where does the pixel-aware mask input come from?
l. 237-239 How does the NTP loss work in this case? Is there causal masking?
l. 252 You're only using the first gripper-close state – might that negatively affect training due to some domain shift between first gripper-close state and subsequent such states?
Section 5.2: Were the $pi_{0}$ and OpenVLA baselines also trained in their typical fashion on the Fractal and Bridge datasets? This would ensure that the improvement is due to the PixelVLA method and instruction tuning approach rather than just seeing more data.

---

> ### Author Response · Authors · 2025-11-22
> **Official Comment by Authors**
>
> We thank the reviewer for recognizing our contributions, particularly (1) the meaningful technical advances in the Pixel-160K dataset and in constructing a VLA that leverages multimodal prompts, and (2) the experiments on two SOTA VLA architectures, which show that PixelVLA can be effectively built on top of multiple types of VLAs.
>
> > **W1:** Unclear whether method is feasible to transfer to novel environments.
>
> **A1:** Thank you for raising this concern. **First**, regarding transfer to novel environments, as shown in Fig. 4, we evaluate PixelVLA and baselines under diverse environment variations on the SimplerEnv-Google Robot setup. The substantial gains of PixelVLA highlight the effectiveness of our visuomotor instruction tuning procedure in handling unseen and novel environments. **Second**, to assess adaptability to new robotic setups, we further adapt PixelVLA to the LIBERO benchmark. The superior results demonstrate that PixelVLA transfers well across embodiments, underscoring the benefit of pixel-level visual understanding and continuous action representation in our framework. We agree that real-world robot experiments would further validate and strengthen the contributions of our approach. We have also made this limitation explicit in Sec. 6 (Limitations) of the revised paper as follows:
>
> “*Although PixelVLA substantially improves VLA performance through pixel-level understanding and tailored visual prompts, it remains limited in handling richer input modalities (e.g., 3D perception) and more advanced forms of visual prompting, such as precise trajectory guidance, reference-image prompts, pose-conditioned prompts, or compositional prompt sequences. In addition, while PixelVLA is extensively validated across three simulated benchmarks, we expect that real-world robot experiments would further strengthen its contributions. We regard these extensions as important directions for future work.*”
>
>
> > **W2:** Missing analysis of varying performance gains and inconsistent performance across tasks
>
> **A2:** Thank you for raising this concern. We have added a clearer analysis and discussion of the varying performance gains and inconsistencies across tasks in the experimental section of the revised paper, and the main changes are summarized below.
>
> 1) “*As shown in Table 3, PixelVLA outperforms OpenVLA across all four LIBERO task suites, with particularly large gains on LIBERO-Long. We attribute this to the Continuous Action Decoder: action chunking helps the policy capture longer-range temporal dependencies [1], while continuous action prediction mitigates compounding discretization errors [2] over long-horizon manipulation.*”
>
> 2) “*Compared to the single-stage Baseline+FT+PUE, PixelVLA adds a first-stage continuous action training, and this two-stage scheme leads to a slight drop on Open/Close Drawer, due to the trade-offs inherent in joint two-stage optimization and the high difficulty and sensitivity of this task.*”
>
> [1] [Bidirectional decoding: Improving action chunking via closed-loop resampling](https://bid-robot.github.io/static/BID_paper.pdf)
>
> [2] [$\pi_0$: A Vision-Language-Action Flow Model for General Robot Control](https://arxiv.org/pdf/2410.24164)
>
> > **Q1:** 218 Where does the pixel-aware mask input come from?
>
> **A1**: Thank you for pointing this out. The pixel-aware mask input is derived from our proposed Pixel-160K dataset using a two-stage annotation pipeline. Specifically, as described in Sec. 3 Problem Definition and Eq. (2), for each training sample {$\mathbf{x}^0, \mathbf{p}^0, \mathbf{a}^0, \mathbf{L}, \mathbf{V}$} drawn from Pixel-160K, we feed the multi-level visual features $\mathbf{F}_v^0$ and the pixel-aware mask $\mathbf{p}^0$ into the multiscale pixel-aware encoder to obtain the pixel-aware embeddings $\mathbf{E}_p^0$. To resolve this confusion, we have added these clarifications to Sec. 4.1 (Multiscale Pixel-aware Encoder) in the revised paper.
>
> > **Q2:** 237-239 How does the NTP loss work in this case? Is there causal masking?
>
> **A2**: We apologize for the confusion. In our current implementation, we do not use a standard autoregressive next-token prediction (NTP) loss, and we do not apply causal masking in the LLM backbone. Instead, the pixel-aware embeddings and other embeddings are jointly fed into the LLM backbone to produce hidden states, which are then passed to the continuous action decoder to generate the continuous action sequence. We supervise this predicted action sequence with an L1 regression loss, as defined in Eq. (4). To resolve this confusion, we have added these clarifications at the end of Sec. 4.1 (Continuous Action Decoder) in the revised paper.

---

> ### Author Response · Authors · 2025-11-22
> **Official Comment by Authors**
>
> > **Q3:** 252 You're only using the first gripper-close state – might that negatively affect training due to some domain shift between first gripper-close state and subsequent such states?
>
> **A3**: Thank you for your question. This design does not negatively affect training due to domain shift, because we only use the first gripper-close state to generate the target object’s region proposal, rather than as a special training state for the policy itself. As illustrated in Step 4 of Fig. 6 in the supplementary material, we make the practical assumption that the first gripper-close state corresponds to the first successful grasp and that the gripper area overlaps with the target object. We then crop around the gripper to obtain an initial region proposal, which greatly facilitates subsequent object segmentation and tracking. This is purely an offline annotation/initialization heuristic and does not restrict the policy to operate only on first gripper-close states, so it does not introduce a harmful domain shift between training and later gripper-close states.
>
> > **Q4:** Concern about the fairness of pi0 and OpenVLA baselines
>
> **A4**: We thank the reviewer for this insightful concern. Both pi0 and OpenVLA in our experiments are initialized from their official checkpoints that are pre-trained on the OXE dataset [3], where the OXE data already includes the Fractal and Bridge datasets. To further ensure fairness and isolate the effect of our PixelVLA design and instruction-tuning strategy, we additionally fine-tune pi0 and OpenVLA on the Fractal and Bridge datasets using exactly the same training configuration as PixelVLA. We denote these stronger baselines as pi0-SFT and OpenVLA-SFT, and have updated Table 1 and Table 2 accordingly in the revised manuscript.
>
> The updated results are summarized in Table 1 below:
> | Methods                          | Pick Coke Can |      | Move Near |      | Open/Close Drawer |      | Average |      |
> |----------------------------------|---------------|------|-----------|------|-------------------|------|---------|------|
> |                                  | VM            | VA   | VM        | VA   | VM                | VA   | VM      | VA   |
> | OpenVLA       | 16.3          | 54.5 | 46.2      | 47.7 | 35.6              | 17.7 | 32.7    | 40.0 |
> | OpenVLA-SFT                      | 17.5          | 51.9 | 44.6      | 42.3 | 32.8              | 16.8 | 31.6    | 38.6 |
> | TraceVLA   | 28.0          | 60.0 | 53.7      | 56.4 | **57.0**          | **31.0** | 46.2 | 49.1 |
> | **PixelVLA**                         | **81.7**      | **72.7** | **60.1** | **57.7** | 42.3      | 20.0 | **61.4** | **50.1** |
> | π₀         | 72.7          | 75.2 | 65.3      | **63.7** | 38.3          | 25.6 | 58.8    | 54.8 |
> | π₀-SFT                           | 70.8          | 72.1 | 64.2      | 61.3 | 36.8              | 28.3 | 57.3    | 53.9 |
> | **PixelVLA-π₀**                      | **80.7**      | **76.8** | **67.7** | 62.0 | **41.3**      | **30.8** | **63.3** | **56.5** |
>
> The updated results are summarized in Table 2 below:
> | Methods                          | Put Spoon |      | Put Carrot |      | Stack Blocks |      | Put Eggplant |      | Average |      |
> |----------------------------------|-----------|------|------------|------|--------------|------|--------------|------|---------|------|
> |                                  | Gra.      | Suc. | Gra.       | Suc. | Gra.         | Suc. | Gra.         | Suc. | Gra.    | Suc. |
> | OpenVLA       | 4.1       | 0.0  | 33.3       | 0.0  | 12.5         | 0.0  | 8.3          | 4.1  | 14.6    | 1.0  |
> | OpenVLA-SFT                      | 8.4       | 0.0  | 35.1       | 12.8 | 10.5         | 0.0  | 16.5         | 8.4  | 17.6    | 5.3  |
> | **PixelVLA**                         | 20.8      | 4.2  | **37.5**       | **20.8** | 16.6         | 0.0  | 79.2         | 41.7 | 38.5    | 16.7 |
> | π₀        | 45.8      | 29.1 | 25.0       | 0.0  | 50.0         | 16.6 | **91.6**         | **62.5** | 53.1    | 27.1 |
> | π₀-SFT                           | 45.3      | 26.8 | 28.6       | 4.2  | 52.3         | 18.6 | 88.5         | 59.6 | 53.7    | 27.3 |
> | **PixelVLA-π₀**                      | **51.7**      | **32.4** | 28.7       | 16.7 | **56.8**         | 21.7 | 83.3         | 61.7 | **55.1**    | **33.8** |
>
> As shown in the updated tables, PixelVLA consistently outperforms both pi0-SFT and OpenVLA-SFT. These results suggest that the incorporation of finer-grained, pixel-level spatial comprehension into existing VLAs enables more effective adaptation to unseen objects and leads to improved out-of-domain generalization.
>
> [3] [Open x-embodiment: Robotic learning datasets and rt-x models: Open x-embodiment collaboration](https://ieeexplore.ieee.org/abstract/document/10611477)

---

> > ### Comment · Reviewer_DBFX · 2025-11-24
> > **Analysis of Results**
> >
> > Thank you to the authors for answering my questions in detail and for including the baseline SFT results in Tables 1 and 2. They have answered the questions I included.
> >
> > Regarding the experiments on synthetic environments (W1), I agree that they adequately show PixelVLA's benefits across multiple environments. I am still a bit concerned about the lack of real world experiments, since it is an important step in robotics research to show applicability in the real world, but since this is not a robotics conference it could be considered a limitation.
> >
> > Regarding W2, do you think that the additional finetuning helps in most tasks but leads to some loss of generalization that could harm performance in, as you put it, highly sensitive and difficult tasks?

---

> > > ### Author Response · Authors · 2025-11-27
> > > **Official Comment by Authors**
> > >
> > > Thank you again for your time and valuable feedback, which has significantly improved our paper. We are glad that our responses and the added SFT baselines in Tables 1 and 2 have addressed most of your concerns, and we appreciate your recognition of both the contributions and limitations of our work.
> > >
> > > > **Q5:** Further discussion about the reason of varying performance gains.
> > >
> > > **A5:** Thanks for the thoughtful follow-up question. Compared with the Baseine+FT+PUE, the additional Continuous Action Training Stage in PixelVLA leads to a slight drop on a few highly sensitive and difficult tasks, which is initially attributed to generic trade-off in joint two-stage optimization. Following your suggestion, we investigate this trade-off more carefully and find that it is fundamentally caused by catastrophic forgetting [4]: the well-known tendency of neural networks. In LLMs, catastrophic forgetting [4] refers to the phenomenon where the model partially or completely loses previously learned abilities (e.g., a slight drop on a few tasks) when it is further fine-tuned on new data or tasks. **However**, compared with the baseline OpenVLA and $\pi_0$, the additional two-stage visuomotor instruction tuning in PixelVLA significantly improves VLA generalization on OOD tasks (Tabs. 1–2), demonstrating that PixelVLA enables more effective generalization to unseen objects and tasks, which provides strong validation for our design.
> > >
> > > We have added this clarification and discussion of catastrophic forgetting in the revised version to make this trade-off more transparent.
> > >
> > > [4] [Revisiting Catastrophic Forgetting in Large Language Model Tuning](https://arxiv.org/pdf/2406.04836)

---

### Official Review · Reviewer_sYs6 · 2025-11-01

**Soundness:** 3
**Presentation:** 3
**Contribution:** 3
**Rating:** 6
**Confidence:** 3

**Summary:**

This paper proposes PixelVLA, a VLA that benefits from pixel-level grounding information. Based on a typical VLA model with a vision encoder, text tokenizer, and LLM, the authors introduced a pixel encoder that is designed to handle pixel-level understanding information. Meanwhile, to finetune such an encoder, the authors proposed a two-stage automated annotation pipeline and created a pixel-annotated visuomotor instruction tuning dataset, Pixel-160k. The experiments on SimplerEnv and LIBERO show that the proposed method achieves better performance than other baselines.

**Strengths:**

The paper is clearly written and well-structured, making it easy to follow. The experiments conducted on SimplerEnv and LIBERO are appropriate and demonstrate the effectiveness of the proposed approach. While introducing an additional pixel-level encoder could intuitively downgrade the pretrained VLM, the authors successfully solve this issue by curating a large 160K dataset and applying LoRA fine-tuning.

**Weaknesses:**

The introduction of pixel-level annotations can be viewed as a relatively straightforward extension of prior work on visual prompting and image-level feature adaptation (e.g., TraceVLA, LLaRA, and related approaches). As a result, the paper’s novelty is somewhat limited. Nonetheless, the work offers useful insights and has potential value for the research community, particularly as a good practice in bridging pixel-level understanding with pretrained VLMs for VLAs.

Meanwhile, the authors are highly encouraged to deploy the proposed method in the real world and verify the claim.

Therefore, I would recommend a weak acceptance.

**Questions:**

N/A

---

> ### Author Response · Authors · 2025-11-22
> **Official Comment by Authors**
>
> We first thank the reviewer for recognizing our contributions, including (1) the clear and well-structured presentation, (2) the effective experiments on SimplerEnv and LIBERO, (3) the value of our Pixel-160K dataset and LoRA fine-tuning in bridging pixel-level understanding with pretrained VLMs for VLAs, and (4) the useful insights and potential value our work provides for the research community.
>
> > **W1:** Comment on novelty and relation to prior visual prompting work.
>
> **A1:** Thank you for this assessment and for acknowledging the potential value of our work. While our visual prompting design is indeed inspired by prior methods (e.g., TraceVLA, LLaRA), our goal is to provide, to the best of our knowledge, the first systematic attempt to enhance VLAs with pixel-level understanding via (i) a multiscale pixel-aware encoder, (ii) an automated pipeline for constructing Pixel-160K with pixel-level annotations, and (iii) a two-stage visuomotor instruction tuning framework. Extensive evaluations on three VLA benchmarks demonstrate that PixelVLA advances current VLAs, achieving superior performance in zero-shot manipulation tasks and better adaptation to new robot setups. We believe the consistent OOD and cross-embodiment gains, achieved with only 1.5% of OpenVLA’s pretraining cost, highlight that this “seemingly straightforward” extension has substantial practical and methodological impact for the community.
>
> In addition, we have expanded the Related Work section with a dedicated paragraph on visual prompting in VLMs, explicitly connecting prior visual-prompt encoding methods to the design of PixelVLA as follows:
>
> “***Visual prompting in VLMs.** Visual prompting methods [1-4] have recently emerged as a complementary paradigm to textual prompting, allowing models to accept more fine-grained supervision in the form of region-level and even pixel-level instructions [5-6] over multimodal inputs. RegionGPT [7] improves region-level understanding in VLMs by enhancing the spatial awareness of visual encoders. Ferret [8] enhances region-level grounding in MLLMs through a hybrid region representation and a spatial-aware visual sampler that supports diverse region inputs, while Ferret-v2 [9] further introduces any-resolution grounding and multi-granularity visual encoding, leading to improved fine-grained visual understanding and localization over prior MLLMs. However, despite these advances, robust pixel-level understanding in VLA frameworks remains challenging, especially when aligning fine-grained spatial cues with continuous, high-precision action control.*”
>
> [1] Vpgtrans: Transfer visual prompt generator across llms.
>
> [2] Pixellm: Pixel reasoning with large multimodal model.
>
> [3] Visionllm v2: An end-to-end generalist multimodal large language model for hundreds of vision-language tasks.
>
> [4] Omg-llava: Bridging image-level, object-level, pixel-level reasoning and understanding.
>
> [5] Groma: Localized visual tokenization for grounding multimodal large language models.
>
> [6] Glamm: Pixel grounding large multimodal model.
>
> [7] Regiongpt: Towards region understanding vision language model.
>
> [8] Ferret: Refer and Ground Anything Anywhere at Any Granularity.
>
> [9] Ferret-v2: An improved baseline for referring and grounding with large language models.
>
>
> > **W2:** Comment on further experimental validation
>
> **A2:** Thank you for your positive recommendation and constructive suggestion. We agree that real-world robot experiments would further validate and strengthen the contributions of our approach. **First**, for transfer to novel environments, Fig. 4 shows that PixelVLA achieves substantial gains over baselines under diverse environment variations in the SimplerEnv-Google Robot setup, demonstrating effective handling of unseen settings. **Second**, for adaptation to new robotic setups, PixelVLA attains superior results on the LIBERO benchmark, indicating good cross-embodiment transfer and highlighting the benefits of pixel-level visual understanding and continuous action representation. Together, these experiments strongly support our claims and provide practical value to the community.
>
> We also discuss this limitation explicit in Sec. 6 (Limitations) of the revised paper as follows:
>
> “*Although PixelVLA substantially improves VLA performance through pixel-level understanding and tailored visual prompts, it remains limited in handling richer input modalities (e.g., 3D perception) and more advanced forms of visual prompting, such as precise trajectory guidance, reference-image prompts, pose-conditioned prompts, or compositional prompt sequences. In addition, while PixelVLA is extensively validated across three simulated benchmarks, we expect that real-world robot experiments would further strengthen its contributions. We regard these extensions as important directions for future work.*”

---

> ### Author Response · Authors · 2025-11-27
> **Official Comment by Authors**
>
> Dear Reviewer sYs6,
>
> Could you please let us know whether your concerns have been fully addressed, as your feedback is very important for improving this work? If any issues remain unclear or if additional suggestions could help strengthen the paper, we would be very grateful to address them and incorporate the changes into the final version. Thank you again for your time and efforts!
>
> Best,
>
> Authors

---

### Official Review · Reviewer_FaYs · 2025-11-01

**Soundness:** 3
**Presentation:** 3
**Contribution:** 2
**Rating:** 2
**Confidence:** 4

**Summary:**

The paper proposes PixelVLA, which besides normal language and , visual encoder, PixelVLA add a modal that enable pixelwise grounding, then claim this enable the model learn more fine-grained represenations that further benefit the understanding and action prediction.

**Strengths:**

1. The paper is well organized and easy to follow.
2. According to the results shown in Table 2, Table 3 and Table 4, it seems, add more pixel level prompting and intergrate this as a new modal can help increase the action control accuracy.

**Weaknesses:**

1. I am confused about the design of the whole architecture. It seems, the author proposes a two-state automated pipeline to get the pixel level segmentation, how these segmentations are used? it is not very clear that which part these segmentation masks are used for in the model.

2. If the segmentation is used as a input to learn the pixel-aware embedding, i am not sure the final optimization of loss for these visual encoder is action accuracy? seems not very relevent. What is the motivation of input these mask to learn the pixel aware embedding.

3. The author gives some results and try to say pixel vla is better than OpenVLA or Pi0, but that is not fair since Pixel VLA needs extra prompting from user, these extra prompt actually gives more direct spatial information of the user's goal. The intuition is the using of these informaiton will give more shortcut to the backbone learning and make it less capable to do reasoning and understand the language.

**Questions:**

Besides the main conern in weakness part. I still have questions:
1. I might need the user better claim the motivation of add the segmentation mask or visual prompting as an input, i feel these will harm the model's potential ability to learn to locate the objects and understanding the scene itself.

---

> ### Author Response · Authors · 2025-11-22
> **Official Comment by Authors**
>
> We first thank the reviewer for recognizing our contributions, noting that (1) the paper is well organized and easy to follow, and (2) the results in Tables 2–4 show that adding pixel-level prompting as an additional modality improves action prediction accuracy.
>
>
> > **W1:** Confusion about the utilization of pixel level segmentation.
>
> **A1:** We are sorry for this confusion. To make the overall architecture clearer, we have redrawn the framework of Fig. 2 and revised on the paper. As shown in the left part of revised Fig. 2 and in Fig. 2(a), the pixel-level segmentation masks are fed into the proposed Multiscale Pixel-aware Encoder, which extracts pixel-aware embeddings. These embeddings are then combined with the visual embeddings, language embeddings, and prompt-aware embeddings, and jointly processed by the LLM to predict the final 7-DoF robot action as shown in Eq. (4). Accordingly, we revise the following description in Sec. 4.1 under Multiscale Pixel-aware Encoder to clarify how the pixel-level masks are fed into the model:
>
> “*Specifically, as described in Eq.(2), for each training sample {$\mathbf{x}^0, \mathbf{p}^0, \mathbf{L}, \mathbf{V}$} drawn from Pixel-160K, PixelVLA first encodes the image observation $\mathbf{x}^0 \in \mathbb{R}^{H \times W \times 3}$ with the SigLIP vision encoder to obtain multi-level visual features $\mathbf{F}^{0}_v$ = {$\mathbf{f}^{0,i}_v, \dots, \mathbf{f}^{0,L}_v$}, where $L$ denotes the number of selected feature levels. As illustrated in Fig.2 (a), the multiscale pixel-aware encoder leverages the features $\mathbf{F}^0_v$ and a pixel-aware mask input $\mathbf{p}^0 \in \mathbb{R}^{H \times W}$ to compute the pixel-aware embeddings $\mathbf{E}_p^0 \in \mathbb{R}^{N_p \times D}$.*”
>
> > **W2:** Questions about the optimization of the visual encoders and the motivation of learning pixel-aware embeddings.
>
> **A2:** Thanks for raising this important point. We address these two questions separately as follows:
>
> 1) Yes, these visual encoders are ultimately optimized using the L1 action prediction loss, as defined in in Eq. (4). In the Pixel-level Understanding Enhancement Stage, we employ LoRA adaptation to efficiently fine-tune PixelVLA’s LLM backbone on the Pixel-160K dataset, while jointly training the visual prompt-aware encoder and the multiscale pixel-aware encoder. All these modules receive gradients purely from the action prediction objective, consistent with the end-to-end visuomotor instruction tuning pipeline.
>
> 2) Following recent advances in equipping MLLMs with region- and pixel-level understanding [1-3], our motivation for introducing segmentation masks is to inject pixel-level feature information into the VLA backbone via a pixel-aware embedding pathway. Specifically, given the multi-level visual features and the pixel-level masks, the proposed Multiscale Pixel-aware Encoder aggregates the masked features into compact pixel-aware embeddings, which are then fed into the LLM backbone. Supervised by the action prediction loss, the model learns to associate the pixel-level information encoded in these pixel-aware embeddings with action generation, thereby enhancing the VLA backbone with pixel-level understanding. We have incorporated the above clarification at the end of Sec. 4.1 under Multiscale Pixel-aware Encoder in the revised paper, explicitly highlighting the motivation behind these pixel-aware embeddings.
>
> [1] [Omg-llava: Bridging image-level, object-level, pixel-level reasoning and understanding](https://proceedings.neurips.cc/paper_files/paper/2024/file/83eb86be3e2f9fd66c44d9073c51ba4d-Paper-Conference.pdf)
>
> [2] [PixelLM: Pixel Reasoning with Large Multimodal Model](https://openaccess.thecvf.com/content/CVPR2024/papers/Ren_PixelLM_Pixel_Reasoning_with_Large_Multimodal_Model_CVPR_2024_paper.pdf)
>
> [3] [VisionLLM v2: An End-to-End Generalist Multimodal Large Language Model for Hundreds of Vision-Language Tasks](https://proceedings.neurips.cc/paper_files/paper/2024/file/81a60d18e010b27b36cd465c6604b915-Paper-Conference.pdf)

---

> ### Author Response · Authors · 2025-11-22
> **Official Comment by Authors**
>
> > **W3:** Concern on fairness and potential weakening of reasoning.
>
> **A3:** Thank you for raising this concern. We address the issues of fairness and the potential weakening of reasoning separately as follows:
>
> 1)	**First**, as detailed in Sec. Experiments, our intention is not to claim that PixelVLA is strictly superior to OpenVLA or Pi0 under extra prompting, but rather to investigate how PixelVLA leverages pixel-level understanding and multimodal prompts to enhance existing VLAs in both in-domain and out-of-domain settings. Hence, we apply the proposed visuomotor instruction tuning framework to two backbone variants, OpenVLA and π0. The results show that PixelVLA consistently outperforms the corresponding baselines in zero-shot manipulation and new-robot adaptation, while using only 1.5% of the pretraining cost of OpenVLA. **Second**, we also compare against TraceVLA, which relies on additional trajectory prompts, and PixelVLA still achieves clearly superior performance. In summary, we believe these comparisons are fair, and the observed improvements provide strong empirical evidence for the contributions of our work.
>
> 2)	**First**, to preserve the general manipulation knowledge—such as reasoning ability and language understanding—learned in the pretrained VLAs, we keep the LLM backbone frozen in both training stages. Benefiting from LoRA adaptation and lightweight attached encoders, the newly introduced modules are plug-and-play and can be easily adapted to various tasks without overwriting the original capabilities. **Second**, as shown in Tabs. 1–2, PixelVLA substantially improves the OOD performance of existing VLAs, indicating that it enhances rather than harms the model’s reasoning ability and multimodal prompt understanding.
>
> > **Q1:** Question about the motivation for adding the segmentation mask and visual prompts as inputs.
>
> **A1:** Thank you for your question. We restate the motivation for introducing segmentation masks and visual prompts as inputs as follows:
>
> 1) Current VLAs largely inherit from MLLMs that process visual inputs only at the image level, without explicit pixel-level understanding. Recent works [1–3] demonstrate that introducing pixel-level understanding (e.g., fine-grained masks as referring inputs) into MLLMs yields substantial gains, as these masks provide precise object representations. This success motivates our first attempt to bring pixel-level understanding into VLAs. By injecting pixel-level information, PixelVLA attains more fine-grained object perception and richer spatial awareness, which is crucial for complex and diverse manipulation environments. As shown in Sec. Experiments, the consistent improvements under OOD evaluation strongly validate this design idea. We have reorganized the part of the Introduction in the revised paper that discusses the motivation for introducing pixel-level understanding.
>
> 2) Visual-prompt-based robotic manipulation has recently attracted substantial attention [4–6], as visual prompts markedly improve performance on unseen tasks and objects while expanding human–robot interaction in OOD scenarios. Therefore, as one of our key contributions, this paper takes an early but important step toward equipping vision–language–action models with robust visual prompting capability.
>
>  [1] [Omg-llava: Bridging image-level, object-level, pixel-level reasoning and understanding](https://proceedings.neurips.cc/paper_files/paper/2024/file/83eb86be3e2f9fd66c44d9073c51ba4d-Paper-Conference.pdf)
>
> [2] [PixelLM: Pixel Reasoning with Large Multimodal Model](https://openaccess.thecvf.com/content/CVPR2024/papers/Ren_PixelLM_Pixel_Reasoning_with_Large_Multimodal_Model_CVPR_2024_paper.pdf)
>
> [3] [VisionLLM v2: An End-to-End Generalist Multimodal Large Language Model for Hundreds of Vision-Language Tasks](https://proceedings.neurips.cc/paper_files/paper/2024/file/81a60d18e010b27b36cd465c6604b915-Paper-Conference.pdf)
>
> [4] [Vima: Robot manipulation with multimodal prompts](https://openreview.net/pdf?id=nkDMZ8yqBt)
>
> [5] [Tracevla: Visual trace prompting enhances spatial-temporal awareness for generalist robotic policies](https://arxiv.org/pdf/2412.10345)
>
> [6] [Object-Centric Prompt-Driven Vision-Language-Action Model for Robotic Manipulation](https://openaccess.thecvf.com/content/CVPR2025/papers/Li_Object-Centric_Prompt-Driven_Vision-Language-Action_Model_for_Robotic_Manipulation_CVPR_2025_paper.pdf)

---

> ### Author Response · Authors · 2025-11-27
> **Official Comment by Authors**
>
> Dear Reviewer FaYs,
>
> Could you please let us know whether your concerns have been fully addressed, as your feedback is very important for improving this work? If any issues remain unclear or if additional suggestions could help strengthen the paper, we would be very grateful to address them and incorporate the changes into the final version. Thank you again for your time and efforts!
>
> Best,
>
> Authors

---

### Author Response · Authors · 2025-11-22
**Official Comment by Authors**

Dear ACs and Reviewers,

We sincerely appreciate the time and effort you have devoted to evaluating our paper. We are encouraged to see that our contributions are consistently recognized across several aspects:

- **Idea.** Reviewers noted that incorporating pixel-level prompting as an additional modality is a simple yet effective idea that improves action control accuracy [Reviewer FaYs and FnpE], and that our method offers a useful practice for bridging pixel-level understanding with pretrained VLMs in VLAs, with meaningful potential for the community [Reviewer sYs6].

- **Experiments.** The experiments on SimplerEnv and LIBERO were considered appropriate and convincing [Reviewer sYs6]. Reviewers highlighted our extensive comparisons against strong SOTA VLAs, including two different architectures, as well as thorough ablations that support our design choices [Reviewer DBFX and FnpE].

- **Dataset & Training Strategy.** The Pixel-160K dataset and our multimodal-prompt VLA design are recognized as meaningful technical contributions [Reviewer DBFX]. Reviewers also noted that the large-scale dataset, LoRA fine-tuning, and our multi-stage training effectively mitigate degradation from the added pixel-level encoder while leveraging existing VLA pretrained weights [Reviewer sYs6 and FnpE].

- **Writing & Presentation.** Multiple reviewers commented that the paper is clearly written, well structured, and easy to follow [Reviewer FaYs and sYs6], which helps in understanding both the method and the empirical results.

And we thank all reviewers for their insightful and constructive suggestions, which greatly helped us improve the paper. In addition to the pointwise responses below, we also summarize the revisions made in the rebuttal according to each reviewer’s comments:

**To Reviewer FaYs:**

1. We redraw Fig. 2 and revise Sec. 4.1 to clearly show how pixel-level masks are encoded into pixel-aware embeddings and optimized with the action prediction loss to inject pixel-level information into the VLA backbone.

2. We clarify that our comparisons are fair, and that PixelVLA improves OOD and cross-robot performance without weakening reasoning ability.

3. We further justify the motivation of using segmentation masks and visual prompts.

**To Reviewer sYs6:**

1. We clarify how PixelVLA differs from prior visual prompting work and emphasize its novelty, and add a dedicated visual prompting paragraph in Related Work to link these methods to our design.

2. We acknowledge the need for further validation, highlight the effectiveness of our current extensive experimental evaluation, and explicitly add a discussion of limitations and future work in Sec. 6.

**To Reviewer DBFX:**

1. We demonstrate that PixelVLA transfers well to novel environments and new robot setups based on our extensive experimental evaluation, and we explicitly add a discussion of limitations and future work in Sec. 6.

2. We add explicit experimental analysis explaining the larger gains on LIBERO-Long and the slight drop on Open/Close Drawer.

3. We clarify the source of the pixel-aware masks and the formulation of the action prediction loss, further explain the motivation for using the first gripper-close state and why it does not introduce domain shift, and additionally fine-tune Pi0 and OpenVLA on Fractal/Bridge as fair comparison baselines.

**To Reviewer FnpE:**

1. We redraw Fig. 2 and revise Sec. 4.1 to clarify the architecture by separating the Visual Prompt-aware Encoder branch, detailing its visual prompt encoder, and explicitly depicting visual prompts as a separate input.

2. We clarify the data generation pipeline by explaining how gripper-based region proposals are formed, specifying that we use Llama 2–7B as the LLM, and describing a two-stage filtering process to remove failed samples.

3. We correct the L307 description to state that only OpenVLA discretizes actions, whereas PixelVLA predicts continuous actions.

4. We add a visual-prompting paragraph to Related Work that links RegionGPT, Ferret, Ferret-v2, etc. to PixelVLA’s design and situates our method within this line of work.

We have highlighted all modifications in the revised paper in blue. We hope these additions address the reviewers’ concerns and further improve our work. If any further clarifications or suggestions would help strengthen the paper, we would be happy to address them and incorporate the changes into the final version. Thank you again for your time and efforts!

Best,

Authors

---

> ### Author Response · Authors · 2025-11-30
> **Summary of Rebuttal Discussion for the Area Chairs**
>
> Dear ACs,
>
> We sincerely appreciate the time and effort you have devoted to our paper. To clarify how we addressed the reviews during the rebuttal discussion, we first briefly summarize the assessments from the Reviewers FnpE, DBFX, and sYs6 who gave a score of 6, and then explain how we resolved the concerns raised by Reviewer FaYs.
>
> **For Reviewer FnpE with a score of 6:**
>
> Overall, [Reviewer FnpE] finds the paper interesting and well-executed, praising its simple yet effective pixel-prompt idea, clever multi-stage training leveraging pretrained VLAs, extensive experiments with thorough ablations. After the rebuttal, [Reviewer FnpE] considers the paper, in its current form, to be strong with most concerns resolved and raises the initial score from 4 to 6, prior to the leakage event on November 28.
>
> **For Reviewer DBFX with a score of 6:**
>
> [Reviewer DBFX] acknowledges the paper’s technical contribution meaningful, highlighting the Pixel-160k dataset and a VLA design that effectively leverages multimodal prompts, and values the evidence of generality across two SOTA VLAs. After the rebuttal, [Reviewer DBFX] praises the detailed responses and the added SFT baselines, noting that the questions have been fully addressed.
>
> **For Reviewer sYs6 with a score of 6:**
>
> [Reviewer sYs6] finds the paper clearly written and well-structured, with appropriately designed experiments that effectively validate the approach. They further conclude that the work provides useful insights  for research community and practical guidance for bridging pixel-level understanding with pretrained VLMs in VLAs.
>
> **For Reviewer FaYs with a score of 2:**
>
> Since [Reviewer FaYs] did not respond during the rebuttal discussion, we rely on our detailed responses and the assessments of the other three reviewers to demonstrate how we have addressed the following main concerns:
>
> 1)  Confusion about the the utilization of pixel level segmentation in architecture.
>
> - To make the overall architecture clearer, we have redrawn the framework in Fig. 2 and updated the corresponding description in the paper. In the revised version (Sec. 4.1), we also provide more detailed explanations of key modules.
> - Notably, this concern was also raised as Weakness 1 by [Reviewer FnpE], who later indicated that it had been resolved by our rebuttal.
>
> 2) Questions about the motivation of learning pixel-aware embeddings and utilizing visual prompts.
>
> - In the revision, we clarified the motivation for introducing pixel-level understanding and visual prompts into VLAs by drawing on their successful use in MLLMs, and we expanded the Introduction and Related Work accordingly.
> - [Reviewer FnpE] praises this “simple yet effective” pixel-prompt idea. Following [Reviewer FnpE]'s suggestion, we now explicitly connect our design to the success of visual prompting in the image domain, further strengthening the motivation.
> - [Reviewer DBFX] views the technical contribution as meaningful, highlighting the Pixel-160k dataset and a VLA design that leverages multimodal prompts and generalizes across multiple VLA backbones.
> - [Reviewer sYs6] notes that the work offers useful insights for the research community and practical guidance for bridging pixel-level understanding with pretrained VLMs in VLAs.
>
> 3) Concern on fairness in experiments and potential weakening of reasoning.
>
> - We clarify that our experimental setup is fair and, in fact, better highlights the core design and main contributions of this work. Moreover, the strong OOD performance shows that PixelVLA preserves, rather than weakens, the generative capabilities of the underlying pretrained model.
> - [Reviewer DBFX] initially questioned the fairness of the OpenVLA and Pi0 baselines but notes that the added SFT results resolve these concerns, and that our experiments on two SOTA VLAs demonstrate PixelVLA can be built on top of multiple VLA backbones.
> - [Reviewer FnpE] highlights that we conduct extensive experiments to demonstrate improvements over SOTA baselines.
> - [Reviewer sYs6] further recognizes that the experiments are appropriate and effectively validate our approach.
>
> We hope these summaries will help the ACs efficiently re-evaluate the contributions of our paper and assess whether the concerns raised by [Reviewer FaYs] have been adequately resolved.
>
> While we are grateful for [Reviewer FaYs]’s efforts, several of the concerns appear to stem from misinterpretations of the problem setting and the proposed method. In our view, they do not point to critical flaws that would justify overlooking the paper’s contributions or assigning a score of 2 leading to rejection. In contrast, three independent reviewers, who engaged closely with the technical core of the work, converged on a positive assessment (score 6). We respectfully hope the ACs will place greater weight on this majority evaluation when making the final decision.
>
> We sincerely appreciate your time and the effort you have devoted to our submission.
>
> Best,
>
> Authors

---

### Meta-Review · Area_Chair_oo9P · 2026-01-05

**Summary:**

The paper introduces PixelVLA, a novel Vision-Language-Action (VLA) model designed to overcome the limitations of current VLAs that primarily rely on image-level understanding and textual instructions. The core contribution is a framework that integrates pixel-level reasoning and multimodal prompting into the robotic control pipeline. To support this, the authors developed Pixel-160K, a large-scale dataset with automated pixel-level annotations derived from existing robot data. During rebuttal, most of the concerns have been carefully addressed. Though the paper has certain limitations, I think it has reached the bar of acceptance.

**Reviewer Concerns:**

Concerns addressed
- Clarity and modality integration. This has been addressed after providing more details
- The novelty against prior visual prompting methods. The authors have added a dedicated subsection discussing this. PixelVLA is the first to bridge multiscale pixel-level understanding specifically with VLA-based robotic control rather than just static VLM tasks
- Other clarifications regarding the data generation pipeline, experiment results, the reason of introducing the segmentation mask etc.

**Reviewer Scores:**

FnpE has raised the score to 6, and sYs6 might raise to 8. Others might just keep the same.

---

### Decision · Program_Chairs · 2026-01-26

Accept (Poster)